# Learning with User-Level Privacy

**Daniel Levy**[*,1]    **Ziteng Sun**[*,2]    **Kareem Amin**[3]    **Satyen Kale**[3]
**Alex Kulesza**[3]    **Mehryar Mohri**[3,4]    **Ananda Theertha Suresh**[3]

[1]Stanford University    [2]Cornell University    [3]Google Research    [4]Courant Institute
danilevy@stanford.edu,    zs335@cornell.edu,
{kamin, satyenkale, kulesza, mohri, theertha}@google.com

## Abstract

We propose and analyze algorithms to solve a range of learning tasks under user-level differential privacy constraints. Rather than guaranteeing only the privacy of individual samples, user-level DP protects a user's entire contribution ($m \geq 1$ samples), providing more stringent but more realistic protection against information leaks. We show that for high-dimensional mean estimation, empirical risk minimization with smooth losses, stochastic convex optimization, and learning hypothesis classes with finite metric entropy, the privacy cost decreases as $O(1/\sqrt{m})$ as users provide more samples. In contrast, when increasing the number of users $n$, the privacy cost decreases at a faster $O(1/n)$ rate. We complement these results with lower bounds showing the minimax optimality of our algorithms for mean estimation and stochastic convex optimization. Our algorithms rely on novel techniques for private mean estimation in arbitrary dimension with error scaling as the concentration radius $\tau$ of the distribution rather than the entire range.

## 1   Introduction

Releasing seemingly innocuous functions of a data set can easily compromise the privacy of individuals, whether the functions are simple counts [35] or complex machine learning models like deep neural networks [52, 30]. To protect against such leaks, Dwork et al. proposed the notion of *differential privacy* (DP). Given some data from $n$ participants in a study, we say that a statistic of the data is differentially private if an attacker who already knows the data of $n - 1$ participants cannot reliably determine from the statistic whether the $n$-th remaining participant is Alice or Bob. With the recent explosion of publicly available data, progress in machine learning, and widespread public release of machine learning models and other statistical inferences, differential privacy has become an important standard and is widely adopted by both industry and government [32, 5, 21, 55].

The standard setting of DP described in [22] assumes that each participant contributes a *single* data point to the dataset, and preserves privacy by "noising" the output in a way that is commensurate with the maximum contribution of a single example. This is not the situation faced in many applications of machine learning models, where users often contribute *multiple* samples to the model—for example, when language and image recognition models are trained on the users' own data, or in federated learning settings [37]. As a result, current techniques either provide privacy guarantees that degrade with a user's increased participation or naively add a substantial amount of noise, relying on the group property of differential privacy, which significantly harms the performance of the deployed model.

To remedy this issue, we consider *user-level* DP, which instead of guaranteeing privacy for individual samples, protects a user's *entire contribution* ($m \geq 1$ samples). This is a more stringent but more realistic privacy desideratum. To hold, it requires that the output of our algorithm does not significantly

---

[*]Equal contribution. Work was done during an internship at Google Research.

35th Conference on Neural Information Processing Systems (NeurIPS 2021).

change when changing user's entire contribution—i.e. possibly swapping up to $m$ samples in total. We make this formal in Definition 1. Very recently, for the reasons outlined above, there has been increasing interest in user-level DP for applications such as estimating discrete distributions under user-level privacy constraints [46], PAC learning with user-level privacy [31], and bounding user contributions in ML models [4, 26]. Differentially private SQL with bounded user contributions was proposed in [59]. User-level privacy has been also studied in the context of learning models via federated learning [49, 48, 58, 6].

In this paper, we tackle the problem of *learning* with user-level privacy in the central model of DP. In particular, we provide algorithms and analyses for the tasks of mean estimation, empirical risk minimization (ERM), stochastic convex optimization (SCO), and learning hypothesis classes with finite metric entropy. Our utility analyses assume that all users draw their samples i.i.d. from related distributions, a setting we refer to as *limited heterogeneity*. On these tasks, naively applying standard mechanisms, such as Laplace or Gaussian, or using the group property with item-level DP estimators, both yield a privacy error independent of $m$. We first develop novel private mean estimators in high dimension with statistical and privacy error scaling with the (arbitrary) concentration radius rather than the range, and apply these to the statistical query setting [SQ; 41]. Our algorithms then rely on (privately) answering a sequence of adaptively chosen queries using users' samples, e.g., gradient queries in stochastic gradient descent algorithms. We show that for these tasks, the additional error due to privacy constraints decreases as $O(1/\sqrt{m})$, contrasting with the naive rate—independent of $m$. Interestingly, increasing $n$, the number of users, decreases the privacy cost at a faster $O(1/n)$ rate.

Importantly, our results imply concrete practical recommendations on sample collection, *regardless of the level of heterogeneity*. Indeed, increasing $m$ will yield the most value in the i.i.d. setting and will yield no improvement when the users' distributions are arbitrary. As the real-world will lie somewhere in between, our results exhibit a regime where, for any heterogeneity, it is strictly better to collect more users (increasing $n$) than more samples per user (increasing $m$).

### 1.1 Our Contributions and Related Work

We provide a theoretical tool to construct estimators for tasks with user-level privacy constraints and apply it to a range of learning problems.

**Optimal private mean estimation and uniformly concentrated queries (Section 3)** We show that for a random variable in $[-B, B]$ concentrated in an unknown interval of radius $\tau$ (made precise in Definition 2), we can privately estimate its mean with error proportional to $\tau$ rather than $B$, as we would obtain using standard private mean estimation techniques such as Laplace mechanism [24]. When data is concentrated in $\ell_\infty$-norm, several papers show that one can achieve an error scaling with $\tau$ rather than $B$, either asymptotically [53], for Gaussian mean-estimation [40, 38], for sub-Gaussian symmetric distributions [18, 17] or for distributions with bounded $p$-th moment [39]. We propose a private mean estimator (Algorithm 2) with error scaling with $\tau$ that works in arbitrary dimension when data is concentrated in $\ell_2$-norm (Theorem 2). In Corollary 1, we show it (optimally) solves mean estimation under user-level privacy constraints for random vectors bounded in $\ell_2$-norm. In Appendix D.6, we show that for uniformly concentrated queries (see Definition 3), sequentially applying Algorithm 2 privately answers $K$ adaptively chosen queries with privacy cost $\tilde{O}(\tau\sqrt{K}/n\varepsilon)$.

Our conclusions relate to the growing literature in adaptive data analysis. While a sequence of work [25, 9, 27, 28] use techniques from differential privacy and their answers are $(\varepsilon, \delta)$-DP with $\varepsilon = \Theta(1)$, our work guarantees privacy for arbitrary $\varepsilon$ with the additional assumption of uniform concentration.

**Empirical risk minimization (Section 4)** An influential line of papers studies ERM under item-level privacy constraints [19, 42, 8]. Importantly, these papers assume *arbitrary* data, i.e., not necessarily samples from users' distributions. The exact analog of ERM in the user-level setting is consequently less interesting as, for $n$ data points $\{z_1, \ldots, z_n\}$, in the worst case, each user $u \in [n]$ contributes $m$ copies of $z_u$ and the problem reduces to the item-level setting. Instead, we consider the (related) problem of ERM when users contribute points sampled i.i.d. Assuming some regularity (A3 and A4), we develop and analyze algorithms for ERM under user-level DP constraints for convex, strongly-convex, and non-convex losses (Theorem 3).

**Optimal stochastic convex optimization (Section 5)** Under item-level DP (or equivalently, user-level DP with $m = 1$), a sequence of work [19, 8, 10, 11, 29] establishes the constrained minimax risk as $\tilde{\Theta}(1/\sqrt{n} + \sqrt{d}/(n\varepsilon))$. In this paper, with the additional assumptions that the losses are

individually smooth[2] and the gradients are sub-Gaussian random vectors, we prove matching upper (Theorem 4) and lower bounds (Theorem 5) of order $\tilde{\Theta}(1/\sqrt{nm} + \sqrt{d}/(n\sqrt{m}\varepsilon))$ in a regime we make precise. We leave closing the gap outside of this regime to future work.

**Limit of learning with a fixed number of users (Appendix B)** Finally, we resolve a conjecture of [4] and prove that with a fixed number of users, even in the limit $m \to \infty$ (i.e., each user has an infinite number of samples), we cannot reach zero error. In particular, we prove that for all the learning tasks we consider, the risk under user-level privacy constraints is at least $\Omega(e^{-\varepsilon n})$ regardless of $m$. Note that this does not contradict the results above since they require $n = \Omega((\log m)/\varepsilon)$.

Finally, we provide results in Appendix A for learning under *pure* user-level DP for function classes with finite metric entropy. We apply these to SCO with $\ell_\infty$ constraints (Remark 1) and achieve (near)-optimal rates.

## 2 Preliminaries

**Notation.** Throughout this work, $d$ denotes the dimension, $n$ the number of users, and $m$ the number of samples per user. Generically, $\sigma$ will denote the sub-Gaussian parameter, $\tau$ the concentration radius, $\nu$ the variance of a random vector and $P$ a data distribution. We denote the optimization variable with $\theta \in \Theta \subset \mathbb{R}^d$, use $z$ (or $Z$ when random) to denote the data sample supported on a space $\mathcal{Z}$, and $\ell \colon \Theta \times \mathcal{Z} \to \mathbb{R}$ for the loss function. Gradients (denoted $\nabla$) are always taken with respect to the optimization variable $\theta$. For a convex set $\mathcal{C}$, $\Pi_\mathcal{C}$ denotes the euclidean projection on $\mathcal{C}$, i.e. $\Pi_\mathcal{C}(y) \coloneqq \operatorname{argmin}_{z \in \mathcal{C}} \|y - z\|_2$. We use A to refer to (possibly random) private mechanisms and $X^n$ as a shorthand for the dataset $(X_1, \ldots, X_n)$. For two distributions $P$ and $Q$, we denote by $\|P - Q\|_{\mathsf{TV}}$ their total variation distance and $D_{\mathsf{kl}}(P\|Q)$ their Kullback-Leibler divergence. For a random vector $X \sim P$ supported on $\mathbb{R}^d$, we use $\mathrm{Var}(P)$ or $\mathrm{Var}(X)$ to denote $\mathbb{E}\left[\|X - \mathbb{E}[X]\|_2^2\right]$, which is equal to the trace of the covariance matrix of $X$.

Next, we consider differential privacy in the most general way, which only requires specifying a dataset space $\mathbb{S}$ and a distance $\mathrm{d}$ on $\mathbb{S}$.

**Definition 1** (Differential Privacy). *Let $\varepsilon, \delta \geq 0$. Let $\mathsf{A} \colon \mathbb{S} \to \Theta$ be a (potentially randomized) mechanism. We say that A is $(\varepsilon, \delta)$-DP with respect to $\mathrm{d}$ if for any measurable subset $O \subset \Theta$ and all $S, S' \in \mathbb{S}$ satisfying $\mathrm{d}(S, S') \leq 1$,*

$$\mathbb{P}(\mathsf{A}(S) \in O) \leq e^\varepsilon \mathbb{P}(\mathsf{A}(S') \in O) + \delta. \tag{1}$$

*If $\delta = 0$, we refer to this guarantee as pure differential privacy.*

For a data space $\mathcal{Z}$, choosing $\mathbb{S} = \mathcal{Z}^n$ and $\mathrm{d}(S, S') = \mathrm{d}_{\mathsf{Ham}}(S, S') = \sum_{i=1}^n 1\{z_i \neq z_i'\}$ recovers the canonical setting considered in most of the literature—we refer to this as *item-level* differential privacy. When we wish to guarantee privacy for *users* rather than individual samples, we instead assume a *structured* dataset into which each of $n$ users contributes $m > 1$ samples. This corresponds to $\mathbb{S} = (\mathcal{Z}^m)^n$ such that for $\mathcal{S} \in \mathbb{S}$, we have

$$\mathcal{S} = (S_1, \ldots, S_n), \text{ where } S_u = \left\{ z_1^{(u)}, \ldots, z_m^{(u)} \right\} \text{ and } \mathrm{d}_{\mathsf{user}}(\mathcal{S}, \mathcal{S}') \coloneqq \sum_{u=1}^n 1\{S_u \neq S_u'\},$$

which means that, in this setting, two datasets are neighboring if at most one of the user's contributions differ. We henceforth refer to this setting as *user-level* differential privacy.

**Distributional assumptions.** In the case of user-level privacy with $n$ users each providing $m$ samples, we assume existence of a collection of distributions $\{P_u\}_{u \in [n]}$ over $\mathcal{Z}$. One then observes the following user-level dataset[3]

$$\mathcal{S} = (S_1, \ldots, S_n) \text{ where } S_u \stackrel{\text{iid}}{\sim} P_u. \tag{2}$$

---

[2]We note that the results only require $\tilde{O}(n^{3/2})$-smooth losses. For large $n$—keeping all other problem parameters fixed—this is a very weak assumption. More precisely, when $n > \operatorname{poly}(d, m, 1/\varepsilon)$, our algorithm on a smoothed version $\tilde{\ell}$ of $\ell$ (e.g., using the Moreau envelope [33]) yields optimal rates for non-smooth losses. Whether the smoothness assumption can be removed altogether is an open question.

[3]For simplicity, we assume that $|S_u| = m$ but our guarantees directly extend to the setting where users have different number of samples with $m$ replaced by $\operatorname{median}(m_1, \ldots, m_n)$ using techniques from [46]. We leave eliciting the optimal rates in settings when $m_u$ is an arbitrary random variable to future work.

In this paper, we consider the *limited heterogeneity* setting, i.e. when the users have related distributions. This setting is more reflective of practice, especially in light of growing interest towards federated learning applications [37, 60].

**Assumption A1** (Limited heterogeneity setting). *There exists a distribution $P_0$ over $\mathcal{Z}$ such that all the user distributions are close to $P_0$ in total variation distance, i.e.*

$$\max_{u \in [n]} \|P_u - P_0\|_{\mathsf{TV}} \leq \Delta,$$

*where $\Delta \geq 0$ quantifies the level of heterogeneity. Note that $\Delta = 0$ corresponds to assumption A2.*

Note that our TV-based definition is natural in this setting as it is closely related to the notion of *discrepancy* (or $d_A$ *distance*) which plays a key role in domain adaption scenarios [47, 12]. Lower bound results have been given in terms of the discrepancy measure (see [13]), which further justify the adoption of this definition in the presence of multiple distributions.

In the case that $\Delta = 0$, A1 reduces to the standard *homogeneous setting*. Many fundamental papers choose this setting when explicating minimax rates under constraints (e.g. in distributed optimization and federated learning [61] or under communication constraints [63, 15]).

**Assumption A2** (Homogeneous setting). *The distributions of individual users are equal, meaning there exists $P_0$ such that for all $u \in [n]$, $P_u = P_0$.*

In this paper, we develop techniques and provide matching upper and lower bounds for solving learning tasks in the homogeneous setting. In Appendix C, we prove that our techniques naturally apply to the heterogeneous setting in a black-box fashion, and for all considered problems provide meaningful guarantees under Assumption A1. Moreover, the algorithm achieves almost optimal rate whenever $\Delta$ is (polynomially) small. See the detailed statement in Theorem 9.

## 2.1 ERM and stochastic convex optimization

**Assumptions on the loss.** Throughout this work, we assume that the parameter space $\Theta$ is closed, convex, and satisfies $\|\theta - \vartheta\|_2 \leq R$ for all $\theta, \vartheta \in \Theta$. We also assume that the loss $\ell \colon \Theta \times \mathcal{Z} \to \mathbb{R}$ is $G$-Lipschitz w.r.t. the $\ell_2$-norm[4], meaning that for all $z \in \mathcal{Z}$, for all $\theta \in \Theta$, $\|\nabla \ell(\theta; z)\|_2 \leq G$. We further consider the following assumptions.

**Assumption A3.** *The function $\ell(\cdot; z)$ is $H$-smooth. In other words, the gradient $\nabla \ell(\theta; z)$ is $H$-Lipschitz in the variable $\theta$ for all $z \in \mathcal{Z}$.*

**Assumption A4.** *The random vector $\nabla \ell(\theta; Z)$ is $\sigma^2$-sub-Gaussian for all $\theta \in \Theta$ and $Z \sim P_0$. Equivalently, for all $v \in \mathbb{R}^d$, $\langle v, \nabla \ell(\theta; Z) \rangle$ is a $\sigma^2$-sub-Gaussian random variable, i.e.,*

$$\mathbb{E}\left[\exp(\langle v, \nabla \ell(\theta; Z) - \mathbb{E}[\nabla \ell(\theta; Z)]\rangle)\right] \leq \exp\left(\|v\|_2^2 \sigma^2 / 2\right).$$

In this work, our rates often depend on the sub-Gaussianity and Lipschitz parameters $\sigma$ and $G$, and thus we define the shorthands $\widetilde{G} := \sigma\sqrt{d}$ and $\underline{G} := \min\{G, \widetilde{G}\}$. Intuitively, the $G$-Lipschitzness assumption bounds the gradient in a ball around 0 (independently of $\theta$), while sub-Gaussianity implies that, for each $\theta$, $\nabla \ell(\theta; Z)$ likely lies in $\mathbb{B}_2^d(\nabla \mathcal{L}(\theta; P_0), \widetilde{G})$. Generically, there is no ordering between $G$ and $\widetilde{G}$: for linear loss $\ell(\theta; z) = \langle \theta, z \rangle$, depending on $P_0$, it can hold that $G \ll \widetilde{G}$ (e.g., $P_0 = \mathsf{Unif}\{-v, v\}$ for $v \in \mathbb{R}^d$), $\widetilde{G} \ll G$ (e.g., $P_0$ is $\mathsf{N}(\mu, \sigma^2 I_d)$ truncated in a ball around $\mu$, with $\|\mu\|_2 \gg \sigma\sqrt{d}$) or $G \approx \widetilde{G}$ (e.g., $P_0 = \mathsf{Unif}\{-1, +1\}^d$).

We introduce the tasks we consider in this work, namely empirical risk minimization (ERM) and stochastic convex optimization (SCO). For a collection of samples from $n$ users $\mathcal{S} = (S_1, \ldots, S_n)$, where each $S_u = \{z_1^{(u)}, \ldots, z_m^{(u)}\} \in \mathcal{Z}^m$, we define the empirical risk objectives

$$\mathcal{L}(\theta; S_u) := \frac{1}{m} \sum_{i=1}^{m} \ell(\theta; z_i^{(u)}) \text{ and } \mathcal{L}(\theta; \mathcal{S}) := \frac{1}{n} \sum_{u=1}^{n} \mathcal{L}(\theta; S_u) = \frac{1}{mn} \sum_{u=1}^{n} \sum_{i=1}^{m} \ell(\theta; z_i^{(u)}). \quad (3)$$

In the user-level setting we wish to minimize $\mathcal{L}(\theta; \mathcal{S})$ under user-level privacy constraints. Going beyond the empirical risk, we also solve SCO [51], i.e. minimizing a convex population objective

---

[4]It is straightforward to develop analogs of the results of Sections 3 and 4 for arbitrary norms, but we restrict our attention to the $\ell_2$ norm in this work for clarity.

when provided with samples from each users' distributions. In the user-level setting, for a convex loss $\ell$ and a convex constraint set $\Theta$, we observe $\mathcal{S} = (S_1, \ldots, S_n) \sim \otimes_{u \in [n]} (P_u)^m$ and wish to

$$\underset{\theta \in \Theta}{\text{minimize}} \ \frac{1}{n} \sum_{u \in [n]} \mathcal{L}(\theta; P_u) \coloneqq \frac{1}{n} \sum_{u \in [n]} \mathbb{E}_{P_u}[\ell(\theta; Z)]. \tag{4}$$

In the homogeneous case (Assumption A2), this reduces to the classic SCO setting:

$$\underset{\theta \in \Theta}{\text{minimize}} \ \mathcal{L}(\theta; P_0) \coloneqq \mathbb{E}_{P_0}[\ell(\theta; Z)]. \tag{5}$$

## 2.2 Uniform concentration of queries

Let $\phi : \mathcal{Z} \to \mathbb{R}^d$ be a $d$-dimensional query function. We define concentration of random variables and uniform concentration of multiple queries as follows.

**Definition 2.** *A (random) sample $X^n$ supported on $[-B, B]^d$ is $(\tau, \gamma)$-concentrated (and we call $\tau$ the "concentration radius") if there exists $x_0 \in [-B, B]^d$ such that with probability at least $1 - \gamma$,*

$$\max_{i \in [n]} \|X_i - x_0\|_2 \leq \tau.$$

**Definition 3** (Uniform concentration of vector queries). *Let $\mathcal{Q}_B^d = \{\phi : \mathcal{Z} \to [-B, B]^d\}$ be a family of queries with bounded range. For $Z^n = (Z_1, \ldots, Z_n) \overset{\text{iid}}{\sim} P$, we say that $(Z^n, \mathcal{Q}_B^d)$ is $(\tau, \gamma)$-uniformly-concentrated if with probability at least $1 - \gamma$, we have*

$$\max_{i \in [n]} \sup_{\phi \in \mathcal{Q}_B^d} \left\| \phi(Z_i) - \mathbb{E}_{Z \sim P}[\phi(Z)] \right\|_2 \leq \tau.$$

In this work, we will often consider $\sigma^2$-sub-Gaussian random variables (or vectors), which are concentrated according to Definition 2. For example, if $X^n$ is drawn i.i.d. from a $\sigma^2$-sub-Gaussian random vector supported on $[-B, B]^d$, then it is $(\sigma \sqrt{d \log(2n/\gamma)}, \gamma)$-concentrated around its mean (see, e.g., [56]). Finally, we define a distance between random variables (and estimators).

**Definition 4** ($\beta$-close Random Variables). *For any two random variables $X_1 \sim P_1$ and $X_2 \sim P_2$, we say $X_1$ and $X_2$ are $\beta$-close, if $\|P_1 - P_2\|_{\text{TV}} \leq \beta$. We use the notation $X_1 \sim_\beta X_2$ if $X_1$ and $X_2$ are $\beta$-close.*

$\beta$-closeness is useful as, in many of our results, the private estimator we propose returns a simple unbiased estimate with high probability and is bounded otherwise. Thus, it suffices to do the analysis in the "nice" case and crudely bound the error otherwise.

## 3 High Dimensional Mean Estimation and Uniformly Concentrated Queries

In this section, we present a private mean estimator with privacy cost proportional to the concentration radius. Using these techniques, we show that, under uniform concentration, we answer adaptively-chosen queries with privacy cost proportional to the concentration radius instead of the whole range. Our theorems guarantee that the estimator is $\beta$-close (with $\beta$ exponentially small in $n$) to a simple unbiased estimator with small noise. We further show how to directly translate these results into bounds on the estimator error, which we demonstrate by providing tight bounds on estimating the mean of $\ell_2$-bounded random vectors under user-level DP constraints (Corollary 1).

Given i.i.d samples $X^n$ from a distribution $P$ supported on $\mathbb{R}^d$ with mean $\mu$, the goal of mean estimation is to design a private estimator that minimizes the $\mathbb{E}\left[\|A(X^n) - \mu\|_2^2\right]$. We focus on distributions with bounded support $[-B, B]^d$. However, our algorithm also generalize to the case when the mean is guaranteed to be in $[-B, B]^d$. In the user-level setting (in the homogeneous case), one observes a dataset $\mathcal{S}$ sampled as in (2) and wishes to minimize $\mathbb{E}[\|A(\mathcal{S}) - \mathbb{E}P_0\|_2^2]$ under user-level privacy constraints. We first focus on the scalar case.

**Mean estimation in one dimension.** The algorithm uses a two-stage procedure, similar in spirit to those of [53], [40], and [39]. In the first stage of this procedure, we use the approximate median estimation in [27], detailed in Algorithm 6 in Appendix D.1, to privately estimate a crude interval

---

**Algorithm 1 WinsorizedMean1D($X^n, \varepsilon, \tau, B$):** Winsorized Mean Estimator (WME)

---

**Require:** $X^n := (X_1, X_2, ..., X_n) \in [-B, B]^n$, $\tau$ : concentration radius, privacy parameter $\varepsilon > 0$.

1: $[a, b] = $ **PrivateRange**$(X^n, \varepsilon/2, \tau, B)$ with $|b - a| = 4\tau$.    {Algorithm 6 in Appendix D.1. }

2: Sample $\xi \sim \text{Lap}\left(0, \frac{8\tau}{\varepsilon n}\right)$ and return

$$\bar{\mu} = \frac{1}{n} \sum_{i=1}^{n} \Pi_{[a,b]}(X_i) + \xi,$$

where $\Pi_{[a,b]}(x) = \max\{a, \min\{x, b\}\}$.

---

in which the means lie, with accuracy $\Theta(\tau)$. The second stage clips the mean around this interval, reducing the sensitivity from $O(B)$ to $O(\tau)$, and adds the appropriate Laplace noise. With high probability, we can recover the guarantee of the Laplace mechanism with smaller sensitivity since the samples are concentrated in a radius $\tau$. We present the formal guarantees of Algorithm 1 in Theorem 1 and defer its proof to Appendix D.2.

**Theorem 1.** *Let $X^n$ be a dataset supported on $[-B, B]$. The output of Algorithm 1, denoted by $\mathsf{A}(X^n)$, is $\varepsilon$-DP. Furthermore, if $X^n$ is $(\tau, \gamma)$-concentrated, it holds that*

$$\mathsf{A}(X^n) \sim_\beta \frac{1}{n} \sum_{i=1}^{n} X_i + \text{Lap}\left(\frac{8\tau}{n\varepsilon}\right),$$

*where $\beta = \min\left\{1, \gamma + \frac{B}{\tau} \exp\left(-\frac{n\varepsilon}{8}\right)\right\}$. Moreover, Algorithm 1 runs in time $\tilde{O}(n + \log(B/\tau))$.*

Compared to [40, 38, 39], our algorithm runs in time $\tilde{O}(n + \log(B/\tau))$ instead of $\tilde{O}(n + B/\tau)$ owing to the approximate median estimation algorithm in [27], which is faster when $\tau \ll B$.

**Mean estimation in arbitrary dimension.** In the general $d$-dimensional case, if $X^n$ is concentrated in $\ell_\infty$-norm, one simply applies Algorithm 1 to each dimension. However, when $X^n$ is concentrated in $\ell_2$-norm, naively upper bounding $\ell_\infty$-norm by the $\ell_2$-norm will incur a superfluous $\sqrt{d}$ factor: if $\|v\|_2 \leq \rho$, each $|v_j|$ is possibly as large as $\rho$. To remedy this issue, we use the random rotation trick in [3, 54]. This guarantees that all coordinates have roughly the same range: for $v \in \mathbb{R}^d$, with high probability, $\|Rv\|_\infty \leq \tilde{O}(\|v\|_2/\sqrt{d})$, where $R$ is the random rotation. We present this procedure in Algorithm 2 and its performance in Theorem 2.

---

**Algorithm 2 WinsorizedMeanHighD($X^n, \varepsilon, \delta, \tau, B, \gamma$):** WME - High Dimension

---

**Require:** $X^n := (X_1, X_2, ..., X_n), X_i \in [-B, B]^d$, $\tau, \gamma$: concentration radius and probability, privacy parameter $\varepsilon, \delta > 0$.

1: Let $D = \text{Diag}(\omega)$ where $\omega$ is sampled uniformly from $\{\pm 1\}^d$.

2: Set $U = d^{-1/2}\mathbf{H}D$, where $\mathbf{H}$ is a $d$-dimensional Hadamard matrix. For all $i \in [n]$, compute
$$Y_i = UX_i.$$

3: Let $\varepsilon' = \frac{\varepsilon}{\sqrt{8d\log(1/\delta)}}, \tau' = 10\tau\sqrt{\frac{\log(dn/\gamma)}{d}}$. For $j \in [d]$, compute
$$\bar{Y}(j) = \textbf{WinsorizedMean1D}\left(\{Y_i(j)\}_{i\in[n]}, \varepsilon', \tau', \sqrt{d}B\right).$$

4: **return** $\bar{X} = U^{-1}\bar{Y}$.

---

**Theorem 2.** *Let $\mathsf{A}(X^n) = \textbf{WinsorizedMeanHighD}(X^n, \varepsilon, \delta, \tau, B, \gamma)$ be the output of Algorithm 2. $\mathsf{A}(X^n)$ is $(\varepsilon, \delta)$-DP. Furthermore, if $X^n$ is $(\tau, \gamma)$-concentrated in $\ell_2$-norm, there exists an estimator $\mathsf{A}'(X^n)$ such that $\mathsf{A}(X^n) \sim_\beta \mathsf{A}'(X^n)$ and*

$$\mathbb{E}[\mathsf{A}'(X^n)|X^n] = \frac{1}{n} \sum_{i=1}^{n} X_i \text{ and } \text{Var}(\mathsf{A}'(X^n)|X^n) \leq c_0 \frac{d\tau^2 \log(dn/\alpha) \log(1/\delta)}{n^2\varepsilon^2}, \qquad (6)$$

*where $c_0 = 102, 400$ and $\beta = \min\left\{1, 2\gamma + \frac{d^2 B \sqrt{\log(dn/\gamma)}}{\tau} \exp\left(-\frac{n\varepsilon}{24\sqrt{d\log(1/\delta)}}\right)\right\}$.*

We present the proof of Theorem 2 in Appendix D.3. We are able to transfer both Theorem 1 and Theorem 2 into finite-sample estimation error bounds for various types of concentrated distributions

and obtain near optimal guarantees (see Appendix D.5 for an example in mean estimation of sub-Gaussian distributions). The next corollary characterizes the risk of mean estimation for distributions supported on an $\ell_2$-bounded domain with user-level DP guarantees (see Appendix D.4 for the proof).

**Corollary 1.** *Assume A2 holds with $P_0$ supported on $\mathbb{B}_2^d(0, B)$ with mean $\mu$. Given $\mathcal{S} = (S_1, S_2, ..., S_n)$, $|S_u| = m$, consisting of $m$ i.i.d. samples from $P_u$. There exists an $(\varepsilon, \delta)$-user-level DP algorithm $\mathsf{A}(\mathcal{S})$ such that, if $n \geq (c_1 \sqrt{d \log(1/\delta)}/\varepsilon) \log(m(dn + n^2\varepsilon^2))$ for a numerical constant $c_1$, we have[5]*

$$\mathbb{E}\left[\|\mathsf{A}(\mathcal{S}) - \mu\|_2^2\right] = \frac{\mathrm{Var}(P_0)}{mn} + \tilde{O}\left(\frac{dB^2}{mn^2\varepsilon^2}\right).$$

*Note that $\mathrm{Var}(P_0) \leq B^2$ for any $P_0$ supported on $\mathbb{B}_2^d(0, B)$. Replacing $\mathrm{Var}(P_0)$ by $B^2$, the bound is minimax optimal up to logarithmic factors. When only A1 holds with $\Delta \leq \mathsf{poly}(d, \frac{1}{n}, \frac{1}{m}, \frac{1}{\varepsilon})$, the same error bounds holds (up to constant) for estimating $\mathbb{E}_{Z \sim P_u}[Z]$ for any $u \in [n]$.*

Note that algorithms in [38, 39], which focus on estimating the mean of $d$-dimensional subGaussian distributions, can also be used to estimate the mean of $\ell_2$-bounded distributions since bounded random variables are also subGaussian. However, applying these algorithms directly will incur a superfluous $d$ factor in the mean square error. We void this using the random rotation trick in Algorithm 2.

**Answering multiple queries.** We end this section by noting that, when a family of queries $\mathcal{Q}$ is uniformly concentrated (as made precise in Definition 3), we answer sequences of $K$ $d$-dimensional, adaptively chosen queries with error scaling as $\tilde{O}(\sqrt{dK}\tau/(n\varepsilon))$ by applying Algorithm 2 to $\{\phi_k(Z_i)\}_{i \in [n]}$ with the right $(\varepsilon_0, \delta_0)$. We make this formal in Theorem 10 in Appendix D.6.

# 4 Empirical Risk Minimization with User-Level Differential Privacy

In this section, we present an algorithm to solve the ERM objective of (3) under user-level DP constraints. We apply the results of Section 3 by noting that the SQ framework encompasses stochastic gradient methods. Informally, one can sequentially choose queries $\phi_k(z) = \nabla\ell(\theta_k; z)$ and, for a stepsize $\eta$, update $\theta_{k+1} = \Pi_\Theta(\theta_k - \eta v_k)$, where $v_k$ is the answer to the $k$-th query. For the results to hold, we require a uniform concentration result over the appropriate class of queries.

**Uniform concentration of stochastic gradients** The class of queries for stochastic gradient methods is $\mathcal{Q}_{\mathsf{erm}} := \{\nabla\ell(\theta; \cdot) : \theta \in \Theta\}$. We prove that when assumptions A3 and A4 hold, $(\{\nabla\ell(\cdot; S_u)\}_{u \in [n]}, \mathcal{Q}_{\mathsf{erm}})$ is $(\tilde{O}(\sigma\sqrt{d/m}), \alpha)$-uniformly concentrated. The next proposition is a simplification of the result of [50] under the (stronger) assumption A3 that $\ell$ is uniformly $H$-smooth. The proof, which we defer to Appendix E.1, hinges on a covering number argument.

**Proposition 1** (Concentration of random gradients). *Let $S_u \overset{\mathrm{iid}}{\sim} P_u$, $|S_u| = m$ for $u \in [n]$ and $\alpha \geq 0$. Under Assumptions A3 and A4, with probability greater than $1 - \alpha$ it holds that*

$$\max_{u \in [n]} \sup_{\theta \in \Theta} \|\nabla\mathcal{L}(\theta; S_u) - \nabla\mathcal{L}(\theta; P_u)\|_2 = O\left(\sigma\sqrt{\frac{d\log\left(\frac{RHm}{d\sigma}\right)}{m} + \frac{\log\left(\frac{n}{\alpha}\right)}{m}}\right).$$

**Stochastic gradient methods** We state classical convergence results for stochastic gradient methods for both convex and non-convex losses under smoothness. For a function $F : \Theta \to \mathbb{R}$, we assume access to a first-order stochastic oracle $\mathsf{O}_{F, \nu^2}$, i.e., a random mapping such that for all $\theta \in \Theta$,

$$\mathsf{O}_{F, \nu^2}(\theta) = \nabla\widehat{F}(\theta) \text{ with } \mathbb{E}\left[\nabla\widehat{F}(\theta)\right] = \nabla F(\theta) \text{ and } \mathrm{Var}\left(\nabla\widehat{F}(\theta)\right) \leq \nu^2.$$

We abstract optimization algorithms in the following way: an algorithm consists of an output set $\mathcal{O}$, a sub-routine $\mathsf{Query} : \mathcal{O} \to \Theta$ that takes the last output and indicates the next point to query and a sub-routine $\mathsf{Update} : \mathcal{O} \times \mathbb{R}^d \to \mathcal{O}$ that takes the previous output and a stochastic gradient and returns the next output. After $T$ steps, we call $\mathsf{Aggregate} : \mathcal{O}^* \to \Theta$, which takes all the previous outputs and returns the final point. (See Algorithm 7 in Appendix E.2 for how to instantiate generic first-order optimization in this framework.) We detail in Proposition 4 in Appendix E.2 standard convergence results for variations of (projected) stochastic gradient descent (SGD). We introduce this abstraction to forego the details of each specific algorithm and instead focus on the privacy and utility guarantees.

---

[5]For precise log factors, see Appendix D.4.

**Algorithm** We recall the ERM setting with user-level DP. We observe $\mathcal{S} = (S_1, \ldots, S_n)$ with $S_u \in \mathcal{Z}^m$ for $u \in [n]$ and wish to solve the constrained optimization problem with objective in (3). We present our method in Algorithm 3 and provide utility and privacy guarantees in Theorem 3.

---

**Algorithm 3** Winsorized First-Order Optimization

---
1: **Input:** Number of iterations $T$, optimization algorithm $\{\mathcal{O}, \mathsf{Query}, \mathsf{Update}, \mathsf{Aggregate}\}$, privacy parameters $(\varepsilon, \delta)$, data $\mathcal{S} = (S_1, \ldots, S_n)$, initial output $o_0$, parameter set $\Theta$, concentration radius $\tau$, probability $\gamma$.
2: Set $\varepsilon' = \frac{\varepsilon}{2\sqrt{2T\log(2/\delta)}}$ and $\delta' = \frac{\delta}{2T}$
3: **for** $t = 0, \ldots, T-1$ **do**
4:     $\theta_t \leftarrow \mathsf{Query}(o_t)$.
5:     For each user $u \in [n]$, compute
$$g_t^{(u)} = \nabla \mathcal{L}(\theta_t; S_u) = \frac{1}{m} \sum_{j \in [m]} \nabla \ell(\theta_t; z_j^{(u)}).$$
6:     Compute $\bar{g}_t = \mathbf{WinsorizedMeanHighD}(\{g_t^{(u)}\}_{u \in [n]}, \varepsilon', \delta', \tau, G, \gamma)$.
7:     $o_{t+1} \leftarrow \mathsf{Update}(o_t, \bar{g}_t)$.
8: **end for**
9: **return** $\bar{\theta} \leftarrow \mathsf{Aggregate}(o_0, \ldots, o_T)$.

---

**Theorem 3** (Privacy and utility guarantees for ERM). *Assume A2 holds and recall that $\widetilde{G} = \sigma\sqrt{d}$, assume[6] $n = \tilde{\Omega}(\sqrt{dT}/\varepsilon)$ and let $\widehat{\theta}$ be the output of Algorithm 3. There exists variants of projected SGD (e.g. the ones we present in Proposition 4) such that, with probability greater than $1 - \gamma$:*

*(i) If for all $z \in \mathcal{Z}, \ell(\cdot; z)$ is convex, then*
$$\mathbb{E}\left[\mathcal{L}(\widehat{\theta}; \mathcal{S}) - \inf_{\theta' \in \Theta} \mathcal{L}(\theta'; \mathcal{S}) \,\Big|\, \mathcal{S}\right] = \tilde{O}\left(\frac{R^2 H}{T} + R\widetilde{G}\frac{\sqrt{d}}{n\sqrt{m}\varepsilon}\right).$$

*(ii) If for all $z \in \mathcal{Z}, \ell(\cdot; z)$ is $\mu$-strongly-convex, then*
$$\mathbb{E}\left[\mathcal{L}(\widehat{\theta}; \mathcal{S}) - \inf_{\theta' \in \Theta} \mathcal{L}(\theta'; \mathcal{S}) \,\Big|\, \mathcal{S}\right] = \tilde{O}\left(GR\exp\left(-\frac{\mu}{H}T\right) + \widetilde{G}^2\frac{d}{\mu n^2 m\varepsilon^2}\right).$$

*(iii) Otherwise, defining the* gradient mapping[7] $\mathsf{G}_{F,\gamma}(\theta) := \frac{1}{\gamma}[\theta - \Pi_\Theta(\theta - \gamma\nabla F(\theta))]$, *we have*
$$\mathbb{E}\left[\|\mathsf{G}_{\mathcal{L}(\cdot;\mathcal{S}),1/H}(\widehat{\theta})\|_2^2|\mathcal{S}\right] = \tilde{O}\left(\frac{H^2 R}{T} + HR\widetilde{G}\frac{\sqrt{d}}{n\sqrt{m}\varepsilon}\right).$$

*For $\varepsilon \leq 1, \delta > 0$, Algorithm 3 instantiated with any first-order gradient algorithm is $(\varepsilon, \delta)$-user-level DP. In the case that only A1 holds, the same guarantees hold whenever $\Delta \leq \mathsf{poly}(d, \frac{1}{n}, \frac{1}{m}, \frac{1}{\varepsilon})$.*

We present the proof in Appendix E.3. For the utility guarantees, the crux of the proof resides in Theorem 10: as well as ensuring small excess loss in expectation, the SQ algorithm produces with high probability a sample from the stochastic gradient oracle $\mathsf{O}_{\mathcal{L}(\cdot;\mathcal{S}),\nu^2}$ where $\nu^2 = \tilde{O}(T\widetilde{G}^2\frac{d}{n^2 m\varepsilon^2})$. When this happens for all $T$ steps, the analysis of stochastic gradient methods provide the desired regret. The privacy guarantees follow from the strong composition theorem of [23].

Importantly, when the function exhibits (some) strong-convexity (which will be the case for any regularized objective), we are able to *localize* the optimal parameter—up to the privacy cost—in $\tilde{O}(H/\mu)$ steps. This will be particularly important in Section 5.

**Corollary 2** (Localization). *Let $\widehat{\theta}$ be the output of Algorithm 3 on the ERM problem of (3). Assume that $\ell(\cdot; z)$ is $\mu$-strongly-convex for all $z \in \mathcal{Z}$, that $n = \tilde{\Omega}(\sqrt{dH/\mu})$ and set $T = \frac{H}{\mu}\log\left(n^2 m(\underline{G}/\widetilde{G}^2)\frac{\mu R\varepsilon^2}{d}\right)$ and $\gamma = \frac{\sigma^2 d^2}{\mu^2 n^2 m\varepsilon^2 R^2}$. For $\theta_{\mathcal{S}}^* \in \mathrm{argmin}_{\theta' \in \Theta} \mathcal{L}(\theta'; \mathcal{S})$, it holds[8]*

---
[6] For precise log factors, see Appendix E.3.

[7] In the unconstrained case—$\Theta = \mathbb{R}^d$—this corresponds to an $\epsilon$-stationary point as $\mathsf{G}_{F,\gamma}(x) = \nabla F(x)$.

[8] A logarithmic dependence on $T$ is hiding in the result. Since $T = \tilde{O}(H/\mu)$, we implicitly assume $H/\mu$ is polynomial in the stated parameters, which is satisfied when we later apply these results to regularized objectives.

$$\mathbb{E}[\|\widehat{\theta} - \theta_{\mathcal{S}}^*\|_2^2] = \tilde{O}\left(\frac{\sigma^2 d^2}{\mu^2 n^2 m \varepsilon^2}\right).$$

# 5  Stochastic Convex Optimization with User-level Privacy

In this section we address the SCO task of (5) under user-level DP constraints. Our approach (which we show in Algorithm 4) solves a sequence of carefully regularized ERM problems, drawing on the guarantees of the previous section. Recall that $\widetilde{G} = \sigma\sqrt{d}$ and $\underline{G} = \min\{G, \widetilde{G}\}$, and that $\ell$ is $H$-smooth under assumption A3. In this section, we assume that $\ell$ is convex. We first present our results and state an upper and lower bound for SCO with user-level privacy constraints.

**Theorem 4** (Phased ERM for SCO). *Algorithm 4 is user-level $(\varepsilon, \delta)$-DP. When A2 holds and $n = \tilde{\Omega}(\min\{\sqrt[3]{d^2 m H^2 R^2/(G\underline{G}\varepsilon^4)}, HR\sqrt{m}/(\sigma\varepsilon)\})$, or, equivalently, $H = \tilde{O}(\sqrt{\frac{n^2\varepsilon^2\sigma^2}{R^2 m} + \frac{G\underline{G}n^3\varepsilon^4}{d^2 R^2 m}})$ for all $P$ and $\ell$ satisfying Assumptions A3 and A4, we have*

$$\mathbb{E}\left[\mathcal{L}(\mathsf{A}_{\mathsf{PhasedERM}}(\mathcal{S}); P_0)\right] - \min_{\theta' \in \Theta} \mathcal{L}(\theta'; P_0) = \tilde{O}\left(\frac{R\sqrt{G\underline{G}}}{\sqrt{mn}} + R\widetilde{G}\frac{\sqrt{d}}{n\sqrt{m}\varepsilon}\right).$$

*Furthermore, our results still hold in the heterogeneous setting (Assumption A1) whenever $\Delta \leq \mathsf{poly}(d, \frac{1}{n}, \frac{1}{m}, \frac{1}{\varepsilon})$; the risk guarantee being with respect to any user distribution $P_u$.*

**Theorem 5** (Lower bound for SCO). *There exists a distribution $P$ and a loss $\ell$ satisfying Assumptions A3 and A4 such that for any algorithm $\mathsf{A}$ satisfying $(\varepsilon, \delta)$-DP at user-level, we have*

$$\mathbb{E}\left[\mathcal{L}(\mathsf{A}(\mathcal{S}); P)\right] - \min_{\theta' \in \Theta} \mathcal{L}(\theta'; P) = \Omega\left(\frac{R\underline{G}}{\sqrt{mn}} + R\underline{G}\frac{\sqrt{d}}{n\sqrt{m}\varepsilon}\right).$$

When $G = \Theta(\sigma\sqrt{d})$, the upper bound matches the lower bound up to logarithmic factors. We present the algorithm and proof for Theorem 4 in Section 5.1. Theorem 5 is proved in Section 5.2.

## 5.1  Upper bound: minimizing a sequence of regularized ERM problems

We now present Algorithm 4, which achieves the upper bound of Theorem 4. It is similar in spirit to Phased ERM [29] and EpochGD [34], in that at each round we minimize a regularized ERM problem with fresh samples and increased regularization, initializing each round from the final iterate of the previous round. This allows us to localize the optimum with exponentially increasing accuracy without blowing up our privacy budget. We solve each round using Algorithm 3 to guarantee privacy and obtain an *approximate* minimizer. We show the guarantee in Corollary 2 is enough to achieve optimal rates. We provide the proof of Theorem 4 in Appendix F and present a sketch here.

---

**Algorithm 4** $\mathsf{A}_{\mathsf{PhasedERM}}$: Phased ERM

---

**Require:** Private dataset: $\mathcal{S} = (S_1, \ldots, S_n) \in (\mathcal{Z}^m)^n : n \times m$ i.i.d samples from $P$, $H$-smooth, convex loss function $\ell$, convex set $\Theta \subset \mathbb{R}^d$, privacy parameters $\varepsilon \leq 1, \delta \leq 1/n^2$, sub-Gaussian parameter $\sigma$.

1: Set $T = \lceil \log_2(\frac{Gn\sqrt{m}\varepsilon}{\sigma d}) \rceil$, $\lambda = \sqrt{\frac{G\underline{G}}{nm} + \frac{\sigma^2 d^2}{n^2 m \varepsilon^2}}/R$
2: **for** $t = 1$ to $T$ **do**
3:     Set $n_t = \frac{n}{2^t}, \lambda_t = 4^t \lambda$.
4:     Sample $\mathcal{S}_t$, $n_t$ users that have not participated in previous rounds. Using Algorithm 3, compute an approximate minimizer $\widehat{\theta}_t$, to the accuracy of Corollary 2, for the objective

$$\mathcal{L}_{\lambda_t, \widehat{\theta}_{t-1}}(\theta; \mathcal{S}_t) = \frac{1}{mn_t}\sum_{u \in \mathcal{S}_t}\sum_{j=1}^{m}\ell(\theta, z_j^{(u)}) + \frac{\lambda_t}{2}\|\theta - \widehat{\theta}_{t-1}\|_2^2. \tag{7}$$

5: **end for**
6: **return** $\widehat{\theta}_T$.

---

*Proof sketch of Theorem 4.* The privacy guarantee comes directly from the privacy guarantee of Algorithm 3 and the fact that $\mathcal{S}_t$ are non-overlapping. The proof for utility is similar to the proof

of Theorem 4.8 in [29]. In round $t$ of Algorithm 4, we consider the true minimizer $\theta_t^*$ and the approximate minimizer $\widehat{\theta}_t$. By stability [14], we can bound the generalization error of $\theta_t^*$ (see Proposition 5 in Appendix F) and, by Corollary 2, we can bound $\mathbb{E}\|\widehat{\theta}_t - \theta_t^*\|_2^2$. We finally choose $\{(\lambda_t, n_t)\}_{t \leq T}$ such that the assumptions of Corollary 2 hold and to minimize the final error. $\qquad \square$

### 5.2  Lower bound: SCO is harder than Gaussian mean estimation

First of all, note that it suffices to prove the lower bounds in the homogeneous setting as any level of heterogeneity only makes the problem harder. Theorem 5 holds for $(\varepsilon, \delta)$-user-level DP—importantly, this is a setting for which lower bounds are generally more challenging (we provide a related lower bound for $\varepsilon$-user-level DP in Appendix A.2). We present the proof in Appendix F.2 and a sketch here.

*Proof sketch of Theorem 5.* The (constrained) minimax lower bound decomposes into a statistical rate and a privacy rate. The statistical rate is optimal (see, e.g., [44, 2]), thus we focus on the privacy rate. We consider linear losses of the form $\ell(\theta; z) = -\langle \theta, z \rangle$. We show that optimizing $\mathcal{L}(\theta; P) = \mathbb{E}_P[\ell(\theta; Z)]$ over $\theta \in \Theta$ is harder than the mean estimation task for $P$. Intuitively, $\mathcal{L}(\theta; P) = -\langle \theta, \mathbb{E}Z \rangle$ attains its minimum at $\theta^* = R\mathbb{E}[Z]/\|\mathbb{E}[Z]\|_2$ and finding $\theta^*$ provides a good estimate of (the direction of) $\mathbb{E}[Z]$. We make this formal in Proposition 6. Next, for Gaussian mean estimation, we reduce, in Proposition 3, user-level DP to item-level DP with lower variance by having each user contribute their sample average (which is a sufficient statistic). We conclude with the results of [38] (see Proposition 7) by proving in Corollary 6 that estimating the direction of the mean with item-level privacy is hard. $\qquad \square$

### Acknowledgments

The authors would like to thank Hilal Asi and Karan Chadha for comments on an earlier draft as well as Yair Carmon, Peter Kairouz, Gautam Kamath, Sai Praneeth Karimireddy, Thomas Steinke and Sebastian Stich, for useful discussions and pointers to very relevant references.

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
