## Discussion

In this work, we explore the fundamental limits of learning under user-level privacy constraints. Importantly, we provide practical algorithms with significantly improved privacy cost in the regime where the number of samples per user $m \gg 1$. However, our work provides generalization guarantees under a limited heterogeneity assumption. Extending our work to more heterogeneous settings is an interesting research direction. Secondly, our work focuses on establishing information-theoretic limits and we do not optimize the runtime of our algorithms. For example, in the case of SCO, our algorithm runs in $\min\{(nm)^{3/2}, n^2 m^{3/2}/\sqrt{d}\}$ time, while achieving the optimal item-level private rate requires at most $\min\{nm, (nm)^2/d\}$ time [10]. Developing faster algorithms in these settings is a possible future direction.

## Potential negative societal impact

Our work is theoretical in nature and we do not foresee major direct negative societal consequences. Because of the growing prevalence of data collection from all sources (mobile, browser, medical records etc.), providing meaningful guarantees—such as user-level DP—while preserving adequate accuracy is an important direction of research. Our work suffers from the same potential negative impact as any work in the broad differential privacy area in two ways: first, a simple way to guarantee privacy is to limit data collection or delete data the users provided in the past. Second, the guarantees we provide are contingent on careful choices of $\varepsilon$ and $\delta$ as well as rigorous and independent methodologies for evaluating the privacy of deployed models.

## A  Function Classes with Bounded Metric Entropy under Pure DP

We consider the general task of learning hypothesis class with finite metric entropy (i.e., such that there exists a finite $\Delta$-cover under a certain norm) and bounded loss under *pure* user-level DP constraints.

For this setting, we present Algorithm 5, which we complement with an information-theoretic lower bound. As in the previous sections, we consider a sample set $\mathcal{S} = (S_1, \ldots, S_n)$, with $S_u = \{z_j^{(u)}\}_{j \in [m]} \subset \mathcal{Z}$. We begin by considering the case of a finite parameter space: for $K \in \mathbb{N}, K < +\infty$, we have

$$\Theta = \left\{ \theta^{(1)}, \ldots, \theta^{(K)} \right\}. \tag{8}$$

For $0 \leq B < \infty$, we denote $\mathcal{F}_B := \{\ell \colon \Theta \times \mathcal{Z} \to \mathbb{R} : \|\ell\|_\infty \leq B\}$ the set of $B$-bounded functions and $\mathcal{A}_\varepsilon^{\mathsf{user}}$ the set $\varepsilon$-user-level DP estimators from $\mathcal{Z}^n$ to $\Theta$, the goal of this section is to elicit the constrained minimax rate [62, 7, 1]

$$\mathfrak{M}_{m,n}^{\mathsf{user}}(\Theta, \mathcal{F}_B, \varepsilon) := \sup_{\mathcal{Z}, \mathcal{P} \subset \mathcal{P}(\mathcal{Z})} \inf_{\mathsf{A} \in \mathcal{A}_\varepsilon^{\mathsf{user}}} \sup_{\ell \in \mathcal{F}_B, P \in \mathcal{P}} \mathbb{E}_{\mathcal{S} \overset{\mathrm{iid}}{\sim} (P^m)^n} \left[ \mathcal{L}(\mathsf{A}(\mathcal{S}); P) - \inf_{\theta \in \Theta} \mathcal{L}(\theta; P) \right].$$

We start with providing the estimator, which combines the private mean estimator of Section 3 with the private selection techniques of [45]. Given a collection of $\varepsilon$-DP mechanisms, the latter provides an $\varepsilon$-DP way to find an (approximate) minimum by sampling from each mechanism at random *with the same data* and returning the maximum of the values observed. In our setup, each mechanism $\mathsf{A}_k$ will be a private release of $\mathcal{L}(\theta^{(k)}; \mathcal{S})$.

### A.1  Combining mean estimation and private selection

Our first step is to show that the conditions of Section 3 are met, that is, the data are concentrated with high probability.

**Lemma 1.** *Let $\mathcal{S} = (S_1, \ldots, S_n) \overset{\mathrm{iid}}{\sim} (P^m)^n$ and $\alpha \in (0, 1)$. With probability greater than $1 - \alpha$, it holds that*

$$\max_{k \in K} \max_{u \in [n]} \left| \mathcal{L}(\theta^{(k)}; S_u) - \mathcal{L}(\theta^{(k)}; P) \right| \leq \frac{B}{2} \sqrt{\frac{\log(|\Theta| \cdot n) + \log(2/\alpha)}{m}}. \tag{9}$$

*In other words, $(\mathcal{S}, \mathcal{Q}_\Theta)$ is $(B/(2\sqrt{m})\sqrt{\log(2Kn/\alpha)}, \alpha)$ uniformly concentrated where $\mathcal{Q}_\Theta = \{\mathcal{L}(\theta; \cdot) : \theta \in \Theta\}$.*

*Proof.* The proof is straightforward: for a fixed $\theta^{(k)} \in \Theta$ and $u \in [n]$, the random variable $\mathcal{L}(\theta^{(k)}; S_u)$ is $\frac{B^2}{4m}$-sub-Gaussian around its mean $\mathcal{L}(\theta^{(k)}; P)$. A union bound over the samples and parameters concludes the proof. $\qquad\square$

Conditioned on that event, the data are well concentrated and the results of Theorem 1 apply. We now describe the algorithm and then go on to prove privacy and utility guarantees. We call it "idealized" because it is not computationally efficient. Roughly, the running time scales as $|\Theta|/\alpha$ to obtain good accuracy with probability greater than $1 - \alpha$. In certain problems, $|\Theta|$ can be exponential in the dimension (e.g., the Lipschitz stochastic optimization problem considered in Remark 1), which makes it computationally intractable.

---

**Algorithm 5** Idealized estimator for learning with bounded losses

1: **Input:** Privacy parameter $\varepsilon$, probability of stopping $\gamma \in (0, 1]$, concentration parameter $\tau > 0$, finite parameter set $\Theta$, dataset $\mathcal{S} = \{S_1, \ldots, S_n\}$
2: Denote
$$\mathsf{A}_k(S) \coloneqq \textbf{WinsorizedMean1D}\Big(\{\mathcal{L}(\theta^{(k)}; S_u)\}_{u \in [n]}, \varepsilon/3, \tau\Big)$$
3: Initialize $\mathcal{T} = \emptyset$.
4: **for** $t = 0, \ldots, \infty$ **do**
5:     Sample $J_t \sim \mathsf{Uniform}(\{1, \ldots, |\Theta|\})$.
6:     Sample $V_t \sim \mathsf{A}_{J_t}(\mathcal{S})$.
7:     Update $\mathcal{T} \to \mathcal{T} \cup \{(J_t, V_t)\}$.
8:     Sample $w_t \sim \mathsf{Bernoulli}(\gamma)$, if $w_t = 1$, break;
9: **end for**
10: $t^* \to \operatorname{argmin}_t V_t$.
11: **return** $(J_{t^*}, V_{t^*})$.

---

We state the privacy and utility of our algorithm. The result follows from the utility guarantees of the mean estimator (Algorithm 1) and the guarantees of private selection in [45].

**Theorem 6.** *Let $\alpha \in (0, 1]$ and let us consider Algorithm 5 with $q = 1/K = 1/|\Theta|$ and $\tau = \frac{B}{2}\sqrt{(\log(Kn) + \log(10/\alpha)/m}$. Assuming that $n \geq \frac{8}{\varepsilon}\log\Big(\frac{25\log(5/\alpha)}{\alpha^2} \cdot \frac{KB}{\tau}\Big)$, the following holds:*

- *(i) The mechanism of Algorithm 5 is $\varepsilon$-user-level DP.*
- *(ii) Let $J_{t^*}$ be the output of Algorithm 5, with probability greater than $1 - \alpha$ it achieves the following utility*

$$\mathcal{L}(\theta^{(J_{t^*})}; \mathcal{S}) - \inf_{\theta' \in \Theta} \mathcal{L}(\theta'; \mathcal{S}) \leq 8 \frac{B}{n\sqrt{m}\varepsilon} \log\Big(25K \cdot \frac{\log(5/\alpha)}{\alpha^2}\Big)\sqrt{\log(Kn) + \log(10/\alpha)}. \tag{10}$$

*Proof.* We first state the privacy guarantee followed by the utility guarantee.

**Proof of (i)** Since each $\mathsf{A}_k$ is $\varepsilon/3$-user-level DP, Theorem 3.2 in [45] guarantees that the output of Algorithm 5 is $\varepsilon$-user-level DP.

**Proof of (ii)** The proof is adapted from Theorem 5.2 in [45]. First of all, with probability greater than $1 - \alpha_1$, as we prove in Lemma 1, the data are uniformly concentrated for all $\theta^{(k)}$, meaning

$$\max_{k \in K} \max_{u \in [n]} \Big|\mathcal{L}(\theta^{(k)}; S_u) - \mathcal{L}(\theta^{(k)}; P)\Big| \leq \left\{\frac{B}{2}\sqrt{\frac{\log(|\Theta| \cdot n) + \log(2/\alpha_1)}{m}} =: \tau\right\}.$$

We condition on this event (Event 1) for the rest of the proof. Let $\alpha_1 \in (0, 1]$ and $\gamma \in (0, 1]$. Let $T_s$ denotes the time that the algortihm exists the loop, which is number of queries the algorithm makes.

Let us denote $k^*$, the best hypothesis in $\Theta$ i.e.

$$k^* = \operatorname*{argmin}_{k \leq K} \mathcal{L}(\theta^{(k)}; \mathcal{S}).$$

We choose $\gamma$ such that $k^*$ is queried with probability greater than $1 - \alpha_1$, i.e., if $E_{\neg k^*}$ is the event (denote $\neg E_{\neg k^*}$ as Event 2) that the algorithm finishes without querying $k^*$, we choose $\gamma$ such that $\mathbb{P}(E_{\neg k^*}) \leq \alpha_1$. More precisely,

$$\begin{aligned}
\mathbb{P}(E_{\neg k^*}) &= \sum_{l=1}^{\infty} \mathbb{P}(E_{\neg k^*}|T_s = l)\mathbb{P}(T_s = l) \\
&= \sum_{l=1}^{\infty} \left(1 - \frac{1}{K}\right)^l \cdot (1-\gamma)^{l-1} \cdot \gamma \\
&= \left(1 - \frac{1}{K}\right)\gamma \sum_{l=0}^{\infty} \left[\left(1 - \frac{1}{K}\right)(1-\gamma)\right]^l \\
&= \frac{\left(1 - \frac{1}{K}\right)\gamma}{1 - \left(1 - \frac{1}{K}\right)(1-\gamma)}.
\end{aligned}$$

Choosing $\gamma = \alpha_1/K$ guarantees that $\mathbb{P}(E_{\neg k^*}) \leq \alpha_1$. Let $L := \frac{\log(1/\alpha_1)}{\gamma} = \log(1/\alpha_1)\frac{K}{\alpha_1}$, we have

$$\mathbb{P}(T_s > L) = \mathbb{P}(\omega_1 = \ldots = \omega_L = 0) = (1-\gamma)^L \leq \exp(-L\gamma) = \alpha_1.$$

Hence with probability at least $1 - \alpha_1$, the algorithm ends in less than $L$ throws (Event 3). Conditioned on this event, by Theorem 1 and union bound, with probability greater than $1 - L \cdot \frac{B}{\tau} \exp(-n\varepsilon/8)$, the output of $\mathsf{A}_{J_t}$ for all $t \leq T_s$ is

$$\mathsf{A}_{J_t}(S) = \mathcal{L}(\theta^{(J_t)}; \mathcal{S}) + \mathsf{Lap}\left(\frac{8\tau}{n\varepsilon}\right) = \frac{1}{m \cdot n} \sum_{j \in [m], u \in [n]} \ell\left(\theta^{(J_t)}; z_j^{(u)}\right) + \mathsf{Lap}\left(\frac{8\tau}{n\varepsilon}\right),$$

which we denote as Event 4. For a Laplace distribution, computing the tail gives that $\mathbb{P}(|\mathsf{Lap}(\lambda)| \geq u) \leq \exp(-u/\lambda)$ and with a union bound and change of variables it holds that if $Y_1, Y_2, \ldots, Y_L \overset{\text{iid}}{\sim} \mathsf{Lap}(\frac{8\tau}{n\varepsilon})$, then with probability greater than $1 - \alpha_1$

$$\max_{i=1,\ldots L} |Y_i| \leq \frac{8\tau}{n\varepsilon} \log\left(\frac{L}{\alpha_1}\right).$$

In other words, except with probability $\alpha_1$, the noise is bounded by $\frac{8\tau}{n\varepsilon} \log(L/\alpha_1)$ (Event 5). Conditioned on all these events, the parameter $\theta^{(J_{t^*})}$ that the algorithm outputs is sub-optimal by at most $\frac{16\tau}{n\varepsilon} \log(L/\alpha_1)$ as in the worst-case the noise is $+\frac{8\tau}{n\varepsilon} \log(L/\alpha_1)$ for $J_{t^*}$ and $-\frac{8\tau}{n\varepsilon} \log(L/\alpha_1)$ for $k^*$. Setting $\alpha_1 = \alpha/5$ and as we assume that $n \geq \frac{8}{\varepsilon} \log\left(\frac{25\log(5/\alpha)}{\alpha^2} \cdot \frac{KB}{\tau}\right)$, we conclude the proof by taking a union bound over all 5 events.

$\square$

**Corollary 3.** *Assume $n \geq \tilde{\Omega}(1)\frac{1}{\varepsilon} \max\left\{\frac{1}{Km}, \log(Km)\right\}$. It holds that*

$$\mathfrak{M}_{m,n}^{\mathsf{user}}(\Theta, \mathcal{F}_B, \epsilon) = \tilde{O}\left(B\left\{\sqrt{\frac{\log K}{m \cdot n}} + \frac{\log^{3/2}(Knm\varepsilon)}{n\sqrt{m}\varepsilon}\right\}\right), \tag{11}$$

*where $\tilde{O}, \tilde{\Omega}$ ignores only numerical constants and log-log factors in this case.*

*Proof.* We get the result directly from Theorem 6, by setting $\alpha = \log K/(n\sqrt{m}\varepsilon)$, applying standard uniform convergence results for bounded losses with finite parameter set (Hoeffding bound) and ignoring log-log factors. $\square$

**Corollary 4** (Parameter sets with finite metric entropy)**.** *Let us further assume that our loss functions are G-Lipschitz with respect to some norm $\|\cdot\|$ with (finite) covering number $\mathsf{N}_{\|\cdot\|}(\Theta, \Delta)$—i.e. there exists a set $\Gamma_{\|\cdot\|,\Delta} \subset \Theta$ such that $|\Gamma_{\|\cdot\|,\Delta}| = \mathsf{N}_{\|\cdot\|}(\Theta, \Delta)$ and for all $\theta \in \Theta$, there exists $\tau \in \Gamma_{\|\cdot\|,\Delta}$*

*such that $\|\theta - \tau\| \leq \Delta$. In this case, for any $\Delta > 0$ and applying Algorithm 5 with parameter set $\Gamma$ guarantees that*

$$\mathfrak{M}_{m,n}^{\mathsf{user}}(\Theta, \mathcal{F}_{B,(G,\|\cdot\|)}, \varepsilon) = \tilde{O}(1) \inf_{\Delta > 0} \left\{ B\left[ \sqrt{\frac{\log \mathsf{N}_{\|\cdot\|}(\Theta, \Delta)}{m \cdot n}} + \frac{\log^{3/2}(\mathsf{N}_{\|\cdot\|}(\Theta, \Delta) n m \varepsilon)}{n\sqrt{m}\varepsilon} \right] + G\Delta \right\}.$$

**Remark 1.** For $\|\cdot\| = \ell_2$, $\Theta = \mathbb{B}_\infty^d(0,1)$ and setting $\Delta = \frac{B}{G}\left\{ \sqrt{d/(mn)} + d^{3/2}/(n\varepsilon\sqrt{m}) \right\}$, we directly get

$$\mathfrak{M}_{m,n}^{\mathsf{user}}(\mathbb{B}_\infty^d(0,1), \mathcal{F}_{B,(G,\ell_2)}, \varepsilon) = \tilde{O}\left\{ B\sqrt{\frac{d}{m \cdot n}} + B\frac{d^{3/2}}{n\sqrt{m}\varepsilon} \right\}.$$

The first term, which corresponds to the statistical rate, is optimal (see e.g. Proposition 2 in [44]). Whether the privacy rate is optimal remains open.

## A.2 Information-theoretic lower bound

We now prove a lower bound on $\mathfrak{M}_{m,n}^{\mathsf{user}}(\Theta, \mathcal{F}_B, \varepsilon)$ when $|\Theta| = K < \infty$. We follow the standard machinery of reducing estimation to testing [62, 57] but under privacy constraints [7, 1].

**Theorem 7** (Lower bound for finite-hypothesis class). *Let $K, m, n \in \mathbb{N}, K < \infty, \varepsilon \in \mathbb{R}_+$, and $0 \leq B < \infty$. Assume $\log_2 K \geq 32 \log 2$ and $n \geq \log_2 K \max\{\frac{1}{192\sqrt{m}\varepsilon}, \frac{1}{96m}\}$, there exists a sample space $\mathcal{Z}$ and parameter set $\Theta$ with $|\Theta| = K$ and $|\mathcal{Z}| = \lceil \log_2 K \rceil$ such that the following holds*

$$\mathfrak{M}_{m,n}^{\mathsf{user}}(\Theta, \mathcal{F}_B, \varepsilon) = \Omega\left( B\sqrt{\frac{\log|\Theta|}{m \cdot n}} + B\frac{\log|\Theta|}{n\sqrt{m}\epsilon} \right). \tag{12}$$

We detail the proof of the theorem below. The proof relies on a (standard) generalization of Fano's method, whcih reduces optimization to multiple hypothesis tests. We refer to the results of [1] to obtain the lower bounds in the case of a constrained—in this case, $\epsilon$-DP—estimators. For the user-level case, we simply consider that samples from an $m$-fold product of measures—the separation does not change but the KL-divergence increase by at most a $m$ factor and TV-distance increase by at most a $\sqrt{m}$ factor thus yielding the final answer.

**Proposition 2** (Acharya et al. [1, Corollary 4]). *Let $\mathcal{P}$ be a collection of distributions over a common sample space $\mathcal{Z}$ and a loss function $\ell : \Theta \times \mathcal{Z} \to \mathbb{R}_+$. For $P, Q \in \mathcal{P}$, define*

$$\mathsf{sep}_{\mathcal{L}}(P, Q; \Theta) := \sup\left\{ \delta \geq 0 \,\middle|\, \text{for all } \theta \in \Theta, \begin{array}{l} \mathcal{L}(\theta, P) \leq \delta \text{ implies } \mathcal{L}(\theta, Q) \geq \delta \\ \mathcal{L}(\theta, Q) \leq \delta \text{ implies } \mathcal{L}(\theta, P) \geq \delta \end{array} \right\}.$$

*Let $\mathcal{V}$ be a finite index set and $\mathcal{P}_{\mathcal{V}} := \{P_v\}_{v \in \mathcal{V}}$ be a collection of distributions contained in $\mathcal{P}$ such that for $\Delta \geq 0$, $\min_{v \neq v'} \mathsf{sep}(P_v, P_{v'}, \Theta) \leq \Delta$. Then*

$$\mathfrak{M}_n^{\mathsf{item}}(\Theta, \mathcal{F}, \epsilon) \geq \frac{\Delta}{4} \max\left\{ 1 - \frac{I(X_1^n; V) + \log 2}{\log|\mathcal{V}|}, \min\left\{ 1, \frac{|\mathcal{V}|}{\exp(c_0 n \varepsilon d_{\mathsf{TV}}(\mathcal{P}_{\mathcal{V}}))} \right\} \right\},$$

*where $V \sim \mathsf{Uniform}(\mathcal{V}), c_0 = 10, d_{\mathsf{TV}}(\mathcal{P}_{\mathcal{V}}) := \max_{v \neq v'} \|P_v - P_{v'}\|_{\mathsf{TV}}$ and $I(X; Y)$ is the (Shannon) mutual information.*

*Proof of Theorem 7.* We follow the standard steps: we first compute the separation, we bound the testing error for any (constrained) estimator in the item-level DP case (with Proposition 2) and finally, we show how to adapt the proof to obtain the user-level DP lower bound.

**Separation** For simplicity, assume $K = 2^d$, if not, the problem is harder than for $\underline{K} = 2^{\lfloor \log_2 K \rfloor} \leq K$ which is of the same order. Let us define the sample space $\mathcal{Z}$, the parameter set $\Theta$ and the loss function $\ell$ we consider.

We define

$$\mathcal{Z} = \Theta := \{-1, +1\}^d \text{ and } \ell(\theta; z) := B\sum_{j \leq d} \mathbf{1}_{\theta_j = z_j}.$$

We consider $\mathcal{V}$ an $d/2$-$\ell_1$ packing of $\{\pm 1\}^d$ of size at least $\exp(d/8)$—which the Gilbert-Varshimov bound (see e.g., [56, Ex. 4.2.16]) guarantees the existence of—and consider the following family of distribution $\mathcal{P} = \{P_v : v \in \mathcal{V}\}$ such that if $X \sim P_v$ then

$$X = \begin{cases} v_j e_j & \text{with probability } \frac{1+\Delta}{2d} \\ -v_j e_j & \text{with probability } \frac{1-\Delta}{2d}. \end{cases} \tag{13}$$

For $\theta \in \Theta$, we have that

$$\mathcal{L}(\theta; P_v) = \mathbb{E}_{P_v}\left[ B \sum_{j \leq d} \mathbf{1}_{\theta_j = Z_j} \right] = B \sum_{j \leq d} \frac{1 + \theta_j v_j \Delta}{2d}.$$

Naturally, $\mathcal{L}(\theta; P_v)$ achieves its minimum at $\theta_v^* = -v$ such that $\inf_{\theta' \in \theta} \mathcal{L}(\theta; P_v) = B\frac{1-\Delta}{2}$. We now compute the separation by noting that

$$\mathsf{sep}_{\mathcal{L}}(P_v, P_{v'}, \Theta) \geq \frac{1}{2} \min_{\theta' \in \Theta} \{ \mathcal{L}(\theta'; P_v) + \mathcal{L}(\theta'; P_{v'}) - \mathcal{L}(\theta_v^*; P_v) - \mathcal{L}(\theta_{v'}^*; P_{v'}) \}. \tag{14}$$

A quick computation shows that $\mathsf{sep}_{\mathcal{L}}(P_v, P_{v'}, \Theta) \geq \frac{B\Delta}{8}$ by noting that $\mathsf{d}_{\mathsf{Ham}}(v, v') \geq d/4$.

**Obtaining the item-level lower bound** We can now use the results of Proposition 2. We have that $\min_{v \neq v'} \mathsf{sep}_{\mathcal{L}}(P_v, P_{v'}, \Theta) \geq \frac{B\Delta}{8}$. The identity $\mathrm{D}_{\mathrm{KL}}(P_v, P_{v'}) = \Delta \log \frac{1+\Delta}{1-\Delta} \leq 3\Delta^2$ implies that $I(Z^n; V) \leq 3n\Delta^2$. Similarly, Pinsker's inequality guarantees that

$$\mathsf{d}_{\mathsf{TV}} \leq \sqrt{\frac{1}{2} \max_{v \neq v'} \mathrm{D}_{\mathrm{KL}}(P_v, P_{v'})} \leq \sqrt{3/2}\Delta.$$

We put everything together and it holds that for $\Delta \in [0, 1]$,

$$\mathfrak{M}_n^{\mathrm{item}}(\Theta, \mathcal{F}, \epsilon) \geq \frac{B\Delta}{32} \max\left\{ 1 - \frac{3n\Delta^2 + \log 2}{d/8}, \min\left\{ 1, \frac{\exp(d/8)}{\exp(30n\varepsilon\Delta)} \right\} \right\}. \tag{15}$$

Since $d \geq 32\log 2$, $\Delta = \sqrt{d/(96n)}$ guarantees that $1 - \frac{3n\Delta^2 + \log 2}{d/8} \geq 1/2$. On the other hand, setting $\Delta = \frac{5}{960}\frac{d}{n\varepsilon}$, guarantees that $\min\left\{ 1, \frac{\exp(d/8)}{\exp(30n\varepsilon\Delta)} \right\} \geq 1/2$. The assumption on $n$ guarantees that these two values are in $[0, 1]$ and thus setting $\Delta^* = \max\left\{ \sqrt{d/(96n)}, \frac{1}{192}\frac{d}{n\varepsilon} \right\}$ which implies that

$$\mathfrak{M}_n^{\mathrm{item}}(\Theta, \mathcal{F}, \epsilon) \geq \frac{B}{32}\left\{ \sqrt{\frac{d}{96n}} + \frac{1}{192}\frac{d}{n\varepsilon} \right\}.$$

**Concluding for user-level DP** Let $m \in \mathbb{N}, m \geq 1$. For the user-level DP lower bound, the proof remains the same except that the collection $\mathcal{P}_{\mathcal{V}}$ becomes $\{P_v^m\}_{v \in \mathcal{V}}$ i.e. the $m$-fold product distribution of $P_v$. The separation remains exactly the same but we now have

$$\mathrm{D}_{\mathrm{KL}}(P_v^m, P_{v'}^m) \leq 3m\Delta^2 \quad \text{and} \quad \mathsf{d}_{\mathsf{TV}}(\mathcal{P}_{\mathcal{V}}) \leq \sqrt{\frac{3m}{2}}\Delta.$$

Under the assumption $\Delta^* = \max\left\{ \sqrt{d/(96mn)}, \frac{1}{192}\frac{d}{n\sqrt{m}\varepsilon} \right\}$ is less than 1 and thus concludes the proof. $\qquad\square$

Note, the upper bound of Theorem 6 and the lower bound above match only up to $\sqrt{\log K}$. Given that $K$ can be exponential in the dimension—e.g. in the case of $\Theta$ being a cover of an $\ell_p$ ball—the bound is only tight for "small" hypothesis class. However, it seems this extra-factor cannot be removed using the techniques we present in this paper, as we need to both obtain uniform concentration and bound the maximum of i.i.d. noise over $K$ samples—both of which are tight. We leave the problem of finding an optimal estimator for this problem to future work.

# B Limit of Learning with a Fixed Number of Users

In this section, we consider the following binary testing problem between $P_1$ and $P_2$ supported on $\{+B, -B\}$ where

$$
\begin{aligned}
P_0(+B) = 1, \quad & P_0(-B) = 0, \\
P_1(+B) = 0, \quad & P_1(-B) = 1.
\end{aligned}
$$

We prove the following result.

**Theorem 8.** *For all user-level $(\varepsilon, \delta)$-DP algorithm $\mathsf{A} : \{+B, -B\}^{m \times n} \to [0, 1]$, let $\mathcal{S}$ be $n \times m$ i.i.d samples from $P_\vartheta, \vartheta \in \{0, 1\}$, we have when $\delta < 1/2ne^{n\varepsilon}$,*

$$
\max_{\theta \in \{0,1\}} \mathbb{E}\left[ (\mathsf{A}(\mathcal{S}) - \vartheta)^2 \right] = \Omega(e^{-n\varepsilon}).
$$

Before proving the theorem, we describe the implications of the theorem to applications considered in this work. Let $\mathcal{A}_{\varepsilon,\delta}^{\mathsf{user}}$ denote the set of all user-level $(\varepsilon, \delta)$-DP algorithms.

**Reduction from mean estimation** $P_0$ and $P_1$ are both bounded distributions. Moreover, we have $\mu_\vartheta = B(2\vartheta - 1)$. For any user-level $(\varepsilon, \delta)$-DP mean estimator $\widehat{\mu} : \{+B, -B\}^{m \times n} \to [-B, +B]$, set $\mathsf{A}_{\widehat{\mu}}(\mathcal{S}) = (\widehat{\mu} + B)/(2B) \in [0, 1]$, we have $\forall \vartheta \in \{0, 1\}$,

$$
\mathbb{E}\left[ (\widehat{\mu}(\mathcal{S}) - \mu_\vartheta)^2 \right] = 4B^2 \mathbb{E}\left[ (\mathsf{A}_{\widehat{\mu}}(\mathcal{S}) - \vartheta)^2 \right].
$$

We have

$$
\inf_{\widehat{\mu} \in \mathcal{A}_{\varepsilon,\delta}^{\mathsf{user}}} \max_{\vartheta \in \{0,1\}} \mathbb{E}\left[ (\widehat{\mu}(\mathcal{S}) - \mu_\vartheta)^2 \right] = 4B^2 \inf_{\widehat{\mu} \in \mathcal{A}_{\varepsilon,\delta}^{\mathsf{user}}} \max_{\vartheta \in \{0,1\}} \mathbb{E}\left[ (\mathsf{A}_{\widehat{\mu}}(\mathcal{S}) - \vartheta)^2 \right]
$$

$$
\geq 4B^2 \inf_{\mathsf{A} \in \mathcal{A}_{\varepsilon,\delta}^{\mathsf{user}}} \max_{\vartheta \in \{0,1\}} \mathbb{E}\left[ (\mathsf{A}(\mathcal{S}) - \vartheta)^2 \right] = \Omega(B^2 e^{-n\varepsilon}).
$$

**Reduction from SCO** Let $\Theta = [-1, 1]$ and $\ell(\theta, Z) = \theta \cdot Z$. Setting $B = G$. The loss is linear (and thus convex), $G$-Lipschitz and satisfies Assumptions A3 and A4. For $P_\vartheta$,

$$
\mathcal{L}(\theta, P_\vartheta) = \theta G(2\vartheta - 1).
$$

Hence the minimizer is $\theta_\vartheta^* = 1 - 2\vartheta$ and

$$
\mathcal{L}(\theta, P_\vartheta) - \mathcal{L}(\theta_\vartheta^*, P_\vartheta) = (2\vartheta - 1)G(\theta - 1 + 2\vartheta) = G(1 - \theta(2\vartheta - 1)) \geq \frac{G}{2}(\theta - 2\vartheta + 1)^2 = \frac{G}{2}(\theta - \mu_\vartheta)^2.
$$

With similar arguments as in the mean estimation reduction, we get

$$
\inf_{\mathsf{A} \in \mathcal{A}_{\varepsilon,\delta}^{\mathsf{user}}} \max_{\vartheta \in \{0,1\}} \mathbb{E}\left[ \mathcal{L}(\mathsf{A}(\mathcal{S}); P_\vartheta) - \min_{\theta \in [-1,1]} \mathcal{L}(\theta; P_\vartheta) \right] = \Omega(Ge^{-n\varepsilon}).
$$

**Reduction from Bounded Losses** In the reduction from SCO, the loss is uniformly bounded and thus this is a sub-problem of the boundeed loss class and the same bound holds.

Finally, let us prove the theorem.

*Proof of Theorem 8.* Note that there is only two possible sets that each user can observe. Let $S_+$ be the multiset consisting of $m$ copies of $+B$ and Let $S_-$ be the multiset consisting of $m$ copies of $-B$. Let $\beta_1 = \mathbb{P}(\mathsf{A}((S_+)^n) < 1/2)$ and $\beta_0 = \mathbb{P}(\mathsf{A}((S_-)^n) \geq 1/2)$. We first show that these two probabilities cannot be simultaneously small.

Since $(S_+)^n$ can be changed into $(S_-)^n$ by changing $n$ users' samples, by group property of differential privacy,

$$
1 - \beta_1 = \mathbb{P}(\mathsf{A}((S_+)^n) \geq 1/2) \leq e^{n\varepsilon} \mathbb{P}(\mathsf{A}((S_-)^n) \geq 1/2) + ne^{n\varepsilon}\delta = e^{n\varepsilon}\beta_0 + ne^{n\varepsilon}\delta.
$$

Similarly, we get

$$
1 - \beta_0 \leq e^{n\varepsilon}\beta_1 + ne^{n\varepsilon}\delta.
$$

Combining the two, we get:

$$\beta_0 + \beta_1 \geq \frac{2(1 - n\delta e^{n\varepsilon})}{1 + e^{n\varepsilon}} \geq \frac{1}{1 + e^{n\varepsilon}}.$$

Note that when $\vartheta = 1$, we have $\mathbb{P}(\mathcal{S} = (S_+)^n) = 1$. Hence

$$\mathbb{E}_{P_1}\left[(\mathsf{A}(\mathcal{S}) - 1)^2\right] \geq \frac{1}{4}\mathbb{P}(\mathsf{A}((S_+)^n) < 1/2).$$

Similarly,

$$\mathbb{E}_{P_0}\left[(\mathsf{A}(\mathcal{S}) - 0)^2\right] \geq \frac{1}{4}\mathbb{P}(\mathsf{A}((S_-)^n) \geq 1/2).$$

We conclude the proof by noting that

$$\max_{\vartheta \in \{0,1\}} \mathbb{E}\left[(\mathsf{A}(\mathcal{S}) - \vartheta)^2\right] \geq \frac{1}{2}\left(\mathbb{E}_{P_0}\left[(\mathsf{A}(\mathcal{S}) - 0)^2\right] + \mathbb{E}_{P_1}\left[(\mathsf{A}(\mathcal{S}) - 1)^2\right]\right).$$

$\square$

## C  Extension to Limited Heterogeneity Setting

In this section, we show that our results and techniques developed under the homogeneous setting (Assumption A2) can be extended to the setting with limited heterogeneity (Assumption A1).

In particular, we show that applying the algorithms under the i.i.d setting in a black-box fashion will work with an additional bounded error under the limited heterogeneity setting, stated in the theorem below.

**Theorem 9.** *Let* $\mathsf{A} : \mathcal{Z}^{m \times n} \to \Theta$ *be a learning algorithm and* $\ell : \mathcal{Z} \times \Theta \to \mathbb{R}_+$ *be a loss function with* $\max_{z \in \mathcal{Z}} \max_{\theta \in \Theta} \mathcal{L}(\theta; z) \leq B$. *Given samples* $\mathcal{S} = (S_1, \ldots, S_n) \sim \otimes_{u \in [n]}(P_u)^m$, *if under Assumption A2, we have*

$$\mathbb{E}\left[\mathcal{L}(\mathsf{A}(\mathcal{S}); P_0)\right] - \min_{\theta' \in \Theta} \mathcal{L}(\theta'; P_0) \leq L(m, n),$$

*then under Assumption A1, we have*

$$\max_{u}\left\{\mathbb{E}\left[\mathcal{L}(\mathsf{A}(\mathcal{S}); P_u)\right] - \min_{\theta' \in \Theta} \mathcal{L}(\theta'; P_u)\right\} \leq L(m, n) + B(mn + 2)\Delta.$$

Before proving the theorem, we can see that for any learning task, when $\Delta < L(m, n)/(B(mn+2))$, we can get the same performance as in the homogeneous case up to constant factors. This is only inverse polynomial in the problem parameters for all considered tasks.

*Proof.* We first show that $\mathcal{S}$ have a similar distribution under Assumption A2 and A1 when $\Delta$ is small. By sub-additivity of total variation distance. Under Assumption A1, we have

$$\|\otimes_{u \in [n]}(P_u)^m - (P_0)^{n \times m}\|_{\mathsf{TV}} \leq mn\Delta. \tag{16}$$

By definition of TV distance, there exists a coupling $(\mathcal{S}, \mathcal{S}')$ where $\mathcal{S} \sim \otimes_{u \in [n]}(P_u)^m$, $\mathcal{S}' \sim (P_0)^{n \times m}$ and

$$\mathbb{P}(\mathcal{S} \neq \mathcal{S}') \leq mn\Delta.$$

Since $\max_{z \in \mathcal{Z}} \max_{\theta \in \Theta} \mathcal{L}(\theta; z) \leq B$, we have

$$\mathbb{E}\left[\mathcal{L}(\mathsf{A}(\mathcal{S}); P_0)\right] - \mathbb{E}\left[\mathcal{L}(\mathsf{A}(\mathcal{S}'); P_0)\right] \leq B \times \mathbb{P}(\mathcal{S} \neq \mathcal{S}') \leq Bmn\Delta. \tag{17}$$

Under Assumption A1, for all $u \in [n]$, $\|P_u - P_0\|_{\mathsf{TV}} \leq \Delta$. For all $\theta \in \Theta$,

$$\mathcal{L}(\theta; P_0) - \mathcal{L}(\theta; P_u) \leq B\Delta,$$

Hence we have

$$\min_{\theta' \in \Theta} \mathcal{L}(\theta'; P_0) - \min_{\theta' \in \Theta} \mathcal{L}(\theta'; P_u) \leq \max_{\theta \in \Theta} |\mathcal{L}(\theta; P_0) - \mathcal{L}(\theta; P_u)| \leq B\Delta, \tag{18}$$

and

$$\mathbb{E}\left[\mathcal{L}(\mathsf{A}(\mathcal{S}'); P_u)\right] - \mathbb{E}\left[\mathcal{L}(\mathsf{A}(\mathcal{S}'); P_0)\right] \leq B\Delta. \tag{19}$$

Therefore, for all $u \in [n]$,

$$\mathbb{E}\left[\mathcal{L}(\mathsf{A}(\mathcal{S}); P_u)\right] - \min_{\theta' \in \Theta} \mathcal{L}(\theta'; P_u)$$

$$= \left(\mathbb{E}\left[\mathcal{L}(\mathsf{A}(\mathcal{S}); P_u)\right] - \mathbb{E}\left[\mathcal{L}(\mathsf{A}(\mathcal{S}'); P_u)\right]\right) + \left(\mathbb{E}\left[\mathcal{L}(\mathsf{A}(\mathcal{S}'); P_u)\right] - \mathbb{E}\left[\mathcal{L}(\mathsf{A}(\mathcal{S}'); P_0)\right]\right)$$

$$+ \left(\mathbb{E}\left[\mathcal{L}(\mathsf{A}(\mathcal{S}'); P_0)\right] - \min_{\theta' \in \Theta} \mathcal{L}(\theta'; P_0)\right) + \left(\min_{\theta' \in \Theta} \mathcal{L}(\theta'; P_0) - \min_{\theta' \in \Theta} \mathcal{L}(\theta'; P_u)\right)$$

$$\leq L(m, n) + B(mn + 2)\Delta,$$

where we bound each term using (16), (17), (18) and (19) respectively. $\qquad\square$

## D  Proofs for Section 3

### D.1  Private range estimation

---
**Algorithm 6 PrivateRange($X^n, \varepsilon, \tau, B$): Private Range Estimation [27]**

---
**Require:** $X^n := (X_1, X_2, ..., X_n) \in [-B, B]^n$, $\tau$ : concentration radius, privacy parameter $\varepsilon > 0$.
1: Divide the interval $[-B, B]$ into $l = B/\tau$ disjoint bins[9], each with width $2\tau$. Let $T$ be the set of middle points of intervals.
2: $\forall i \in [n]$, let $X_i' = \min_{x \in T} |X_i - x|$ be the point in $T$ closest to $X_i$.
3: $\forall x \in T$, define cost function

$$c(x) = \max\{|\{i \in [n] \mid X_i' < x\}|, |\{i \in [n] \mid X_i' > x\}|\}.$$

4: Sample $x \in T$ based on the following distribution:

$$\mathbb{P}(\hat{\mu} = x) = \frac{e^{-\varepsilon c(x)/2}}{\sum_{x' \in T} e^{-\varepsilon c(x')/2}}.$$

5: Return $R = [\hat{\mu} - 2\tau, \hat{\mu} + 2\tau]$.

---

### D.2  Proof of Theorem 1

**Theorem 1.** *Let $X^n$ be a dataset supported on $[-B, B]$. The output of Algorithm 1, denoted by $\mathsf{A}(X^n)$, is $\varepsilon$-DP. Furthermore, if $X^n$ is $(\tau, \gamma)$-concentrated, it holds that*

$$\mathsf{A}(X^n) \sim_\beta \frac{1}{n} \sum_{i=1}^{n} X_i + Lap\left(\frac{8\tau}{n\varepsilon}\right),$$

*where $\beta = \min\left\{1, \gamma + \frac{B}{\tau} \exp\left(-\frac{n\varepsilon}{8}\right)\right\}$. Moreover, Algorithm 1 runs in time $\tilde{O}(n + \log(B/\tau))$.*

*Proof.* The privacy guarantee of the algorithm follows from the composition theorem of DP and the privacy guarantees of the exponential and Laplace mechanisms. For utility, it is enough to show that with probability at least $1 - (\gamma + \frac{B}{\tau} \exp\left(-\frac{n\varepsilon}{8}\right))$, $\forall i \in [n], X_i$ is not truncated, i.e. $X_i \in [\hat{\mu} - 2\tau, \hat{\mu} + 2\tau]$.

Recall that $X_i'$ is the middle of the interval in which $X_i$ falls. By the definition of $(\tau, \gamma)$-concentration, with probability at least $1 - \gamma, \forall i \in [n]$,

$$|X_i - x_0| \leq \tau.$$

This implies that $\forall i \in [n]$,

$$|X_i' - x_0| \leq 2\tau,$$

hence so is the $\left(\frac{1}{4}, \frac{3}{4}\right)$-quantile of $\{X_i'\}_{i=1}^n$. According to [27] (Theorem 3.1), Algorithm 6 outputs $\left(\frac{1}{4}, \frac{3}{4}\right)$-quantile of $\{X_i'\}_{i=1}^n$ with probability at least $1 - \frac{B}{\tau} e^{-\frac{n\varepsilon}{8}}$. The proof follows by a union bound of both events. $\qquad\square$

---
[9]The last interval is of length $2B - (t-1)\tau$ if $\tau$ doesn't divide $B$.

## D.3 Proof of Theorem 2

**Theorem 2.** *Let* $\mathsf{A}(X^n) = \textbf{\textit{WinsorizedMeanHighD}}(X^n, \varepsilon, \delta, \tau, B, \gamma)$ *be the output of Algorithm 2.* $\mathsf{A}(X^n)$ *is* $(\varepsilon, \delta)$-*DP. Furthermore, if* $X^n$ *is* $(\tau, \gamma)$-*concentrated in* $\ell_2$-*norm, there exists an estimator* $\mathsf{A}'(X^n)$ *such that* $\mathsf{A}(X^n) \sim_\beta \mathsf{A}'(X^n)$ *and*

$$\mathbb{E}[\mathsf{A}'(X^n)|X^n] = \frac{1}{n}\sum_{i=1}^{n} X_i \ \text{ and } \ \mathrm{Var}(\mathsf{A}'(X^n)|X^n) \leq c_0 \frac{d\tau^2 \log(dn/\alpha)\log(1/\delta)}{n^2\varepsilon^2}, \qquad (6)$$

*where* $c_0 = 102,400$ *and* $\beta = \min\left\{1, 2\gamma + \frac{d^2 B\sqrt{\log(dn/\gamma)}}{\tau}\exp\left(-\frac{n\varepsilon}{24\sqrt{d\log(1/\delta)}}\right)\right\}$.

We start by proving the following Lemma, which states that if the data is concentrated in $\ell_2$-norm with radius $\tau$, then after a random rotation, the points are concentrated in $\ell_\infty$-norm with radius $\tau/\sqrt{d}$ up to logarithmic factors.

**Lemma 2.** *Let* $U = \frac{1}{\sqrt{d}}\mathbf{H}D$, *where* $\mathbf{H}$ *is the Walsh Hadamard matrix and* $D$ *is a diagonal matrix with i.i.d. uniformly random* $\{+1, -1\}$ *entries. Let* $x_1, x_2, \ldots, x_n$ *and* $x_0$ *be vectors in* $\mathbb{R}^d$. *With probability at least* $1 - \alpha$, *then the following holds.*

$$\max_i \|Ux_i - Ux_0\|_\infty \leq \frac{10\max_i \|x_i - x_0\|_2 \sqrt{\log\frac{nd}{\alpha}}}{\sqrt{d}}.$$

*Proof.* Let $z_i = x_i - x_0$. It suffices to show that

$$\max_i \|Uz_i\|_\infty \leq \frac{10\max_i \|z_i\|_2 \sqrt{\log\frac{nd}{\alpha}}}{\sqrt{d}}.$$

holds with probability at least $1 - \alpha$. Let $y_i = Uz_i$ and let $y_{i,j}$ denote the $j^{\text{th}}$ coordinate of $y_j$. Let $D_j$ denote that $j^{th}$ diagonal of $D$. Then

$$y_{i,j} = \frac{1}{\sqrt{d}}\sum_k \mathbf{H}_{j,k}D_k z_{i,k}$$

Hence,

$$\mathbb{E}[y_{i,j}] = \frac{1}{\sqrt{d}}\sum_k \mathbf{H}_{j,k}\mathbb{E}[D_k]z_{i,k} = 0.$$

However, observe that changing one coordinate of $D$, say $D_k$ changes the value of $y_{i,j}$ by at most

$$y_{i,j} - y'_{i,j} \leq \frac{2}{\sqrt{d}}z_{i,k} \leq \frac{2\|z_i\|_2}{\sqrt{d}}.$$

Hence, by the McDiarmid's inequality with probability at least $1 - \alpha'$

$$|y_{i,j}| \leq \frac{10\|z_i\|_2 \sqrt{\log\frac{1}{\alpha'}}}{\sqrt{d}}.$$

Choosing $\alpha' = \alpha/nd$ and applying union bound over all coordinates of all vectors yields the desired bound. $\qquad\square$

Thus, after applying the random rotation, we have with probability $1 - 2\gamma$ that for all $j \in [d]$, $\{Y_i(j)\}_{u\in[n]}$ is $(\tau', 0)$-concentrated with $\tau' = 10\tau\sqrt{\log(nd/\alpha)/d}$. Hence conditioned on this event, by Theorem 1 and a union bound over $d$ coordinates, after applying **WinsorizedMean1D** to each dimension, we have that for all $j \in [d], \bar{Y}(j) \sim_\beta \bar{Y}'(j)$ where $\beta = 1 - \frac{\sqrt{d}B}{\tau'}\exp\left(-\frac{n\varepsilon'}{8}\right)$ and

$$\bar{Y}'(j) = \frac{1}{n}\sum_{i=1}^n Y_i(j) + \mathrm{Lap}\left(\frac{8\tau'}{n\varepsilon'}\right),$$

Plugging in values of $\tau'$ and $\varepsilon'$, it can be seen that $\bar{Y}'$ satisfies the conditions in the theorem. By subadditivity of TV distance, we have

$$\bar{Y} \sim_{d\beta} \bar{Y}'.$$

The theorem follows by noting the random rotation is an orthogonal transform and preserves variance.

**D.4 Proof of Corollary 1**

For all $i \in [n]$, let $X_i = \frac{1}{m} \sum_{j=1}^{m} Z_j^{(i)}$, i.e., the average of user $i$'s samples. Since $\|Z_j^{(i)}\| \leq B$, we know that $X^n$ is $(B\sqrt{\log(2n/\gamma)/(2m)}, \gamma)$-concentrated (e.g., see [36]). Hence by Theorem 2, if we apply Algorithm 2 to $X^n$, we have $\mathsf{A}(X^n) \sim_\beta \mathsf{A}'(X^n)$ with $\beta = \min\{1, \gamma + \alpha + \frac{d^2 B \sqrt{\log(dn/\alpha)}}{\tau} \exp(-\frac{n\varepsilon}{24\sqrt{d\log(1/\delta)}})\}$ with $\tau = B\sqrt{\log(2n/\gamma)/(2m)}$ and

$$\mathbb{E}[\mathsf{A}'(X^n)|X^n] = \frac{1}{n} \sum_{i=1}^{n} X_i \text{ and } \mathrm{Var}(\mathsf{A}'(X^n)|X^n) \leq c_0 \frac{d\tau^2 \log(dn/\alpha)\log(1/\delta)}{n^2\varepsilon^2}.$$

Hence

$$\mathbb{E}[\mathsf{A}'(X^n)] = \mathbb{E}[\mathbb{E}[\mathsf{A}'(X^n)|X^n]] = \mathbb{E}\left[\frac{1}{n}\sum_{i=1}^{n} X_i\right] = \mu.$$

$$\begin{aligned}
\mathrm{Var}(\mathsf{A}'(X^n)) &= \mathbb{E}[\mathrm{Var}(\mathsf{A}'(X^n)|X^n)] + \mathrm{Var}(\mathbb{E}[\mathsf{A}'(X^n)|X^n]) \\
&\leq \mathrm{Var}\left(\frac{1}{n}\sum_{i=1}^{n} X_i\right) + c_0 \frac{d\tau^2 \log(dn/\alpha)\log(1/\delta)}{n^2\varepsilon^2} \\
&= \frac{\mathrm{Var}(P_0)}{mn} + c_0 \frac{dB^2 \log(2n/\gamma)\log(dn/\alpha)\log(1/\delta)}{mn^2\varepsilon^2}.
\end{aligned}$$

Combining the two, we have

$$\mathbb{E}\left[\|\mathsf{A}'(X^n) - \mu\|_2^2\right] \leq \frac{\mathrm{Var}(P_0)}{mn} + c_0 \frac{dB^2 \log(2n/\gamma)\log(dn/\alpha)\log(1/\delta)}{mn^2\varepsilon^2}.$$

Since $\mathsf{A}(X^n) \sim_\beta \mathsf{A}'(X^n)$, we have

$$\mathbb{E}\left[\|\mathsf{A}(X^n) - \mu\|_2^2\right] \leq \frac{\mathrm{Var}(P_0)}{mn} + c_0 \frac{dB^2 \log(2n/\gamma)\log(dn/\alpha)\log(1/\delta)}{mn^2\varepsilon^2} + \beta B^2.$$

Taking $\alpha = \gamma = \frac{c_0 d}{3mn^2\varepsilon^2}$, we have when $n \geq c_1 \frac{\sqrt{d\log(1/\delta)}}{\varepsilon} \log(dm^{3/2}\varepsilon^2)$ for a constant $c_1$, we have

$$\mathbb{E}\left[\|\mathsf{A}(X^n) - \mu\|_2^2\right] \leq \frac{\mathrm{Var}(P_0)}{mn} + c_0 \frac{2dB^2 \log(mn^2\varepsilon^2/d)\log(mn^3\varepsilon^2))\log(1/\delta)}{mn^2\varepsilon^2}.$$

**Tightness of Corollary 1.** The first term is the classic statistical rate even with unconstrained access to the samples. We prove the tightness of the second term using the following family of truncated Gaussian distributions. The proof follows a similar line of argument of the proof for Theorem 5 in Section F.2. For a mean $\mu \in \mathbb{R}^d$, a covariance $\Sigma \in \mathbb{R}^{d \times d}$ and $B > 0$, we consider the family of $\ell_\infty$-truncated Gaussians, meaning

$$Z \sim \mathsf{N}^{\mathrm{tr}}(\mu, \Sigma, B) \text{ if } Z_0 \sim \mathsf{N}(\mu, \Sigma) \text{ and set for all } j \in [d] \ Z(j) = \frac{Z_0(j)}{\max\{1, |Z_0(j)|/B\}}. \quad (20)$$

In other words, the standard high-dimensional Gaussian distribution where the mass outside of $\mathbb{B}_\infty^d(0, B)$ has been projected back onto the hyperrectangle coordinate-wise.

In this proof, we will take $\Sigma = \sigma^2 I_d$. We first state the following Lemma, proved in Section F.2, which shows that when $B$ is large enough compared to $\|\mu\|_2$ and $\sigma$, then the expectation of $\mathsf{N}^{\mathrm{tr}}(\mu, \sigma^2 I_d, B/\sqrt{d})$ and $\mu$ are exponentially close in $\ell_2$-norm.

**Lemma 3.** *Suppose* $\|\mu\|_2 + 10\sqrt{d}\sigma < G,$

$$\|\mathbb{E}_{Z \sim \mathsf{N}^{\mathrm{tr}}(\mu, \sigma^2 I_d, G/\sqrt{d})}[Z] - \mu\|_2 = O(\sigma e^{-10d}).$$

**Reducing to standard Gaussian mean estimation** We will take $\sigma = B/20\sqrt{d}$ and $\|\mu\|_2 \leq B/2$, Hence assuming $m, n$ is polynomial in $d$, $O(\sigma e^{-10d})$ is small compared to the bound in Corollary 1.

Note that we can always simulate a sample from $\mathsf{N}^{\mathrm{tr}}(\mu, \sigma^2 I_d, B/\sqrt{d})$ using a sample from $\mathsf{N}(\mu, \sigma^2 I_d)$ by performing truncation. Taking $\sigma = B/20\sqrt{d}$, it would be enough to prove the following:

$$\inf_{\widehat{\mu} \in \mathcal{A}^{\mathrm{item}}_{\varepsilon, \delta}} \sup_{\mu: \|\mu\|_2 \leq B/2} \mathbb{E}_{\mathcal{S} \overset{\mathrm{iid}}{\sim} \mathsf{N}(\mu, \sigma^2 I_d)} \left[\|\widehat{\mu}(\mathcal{S}) - \mu\|_2^2\right] = \tilde{\Omega}\left(\frac{d^2 \sigma^2}{mn^2 \varepsilon^2}\right),$$

where $\mathcal{A}^{\mathrm{user}}_{\varepsilon, \delta}$ denotes set of all user-level $(\varepsilon, \delta)$-DP algorithms. The next proposition, based on the fact that sample mean is a sufficient statistic for i.i.d Gaussian samples, shows that we can reduce the problem to Gaussian mean estimation under item-level DP, with a smaller variance. The proposition is proved in Section F.2.

**Proposition 3** (From multiple samples to one good sample)**.** *Suppose each user $u \in [n]$ observe* $(Z_1^{(u)}, \ldots, Z_m^{(u)}) \overset{\mathrm{iid}}{\sim} \mathsf{N}(\mu, \sigma^2 I_d)$. *For any $(\varepsilon, \delta)$ user-level DP algorithm $\mathsf{A}^{\mathrm{user}}$, there exists an $(\varepsilon, \delta)$-item-level DP algorithm $\mathsf{A}^{\mathrm{item}}$ that takes as input $(\bar{Z}^{(1)}, \ldots, \bar{Z}^{(n)})$ with $\bar{Z}^{(u)} := \frac{1}{m} \sum_{j \leq m} Z_j^{(u)}$ and has the same performance as $\mathsf{A}^{\mathrm{user}}$.*

Since $\bar{Z}^{(u)}$ is a sample from $\mathsf{N}(\mu, \frac{\sigma^2}{m} I_d)$, it remains to prove

$$\inf_{\widehat{\mu} \in \mathcal{A}^{\mathrm{item}}_{\varepsilon, \delta}} \sup_{\mu: \|\mu\|_2 \leq B/2} \mathbb{E}_{Z^n \overset{\mathrm{iid}}{\sim} \mathsf{N}(\mu, \frac{\sigma^2}{m} I_d)} \left[\|\widehat{\mu}(Z^n) - \mu\|_2^2\right] = \tilde{\Omega}\left(\frac{d^2 \sigma^2}{mn^2 \varepsilon^2}\right),$$

where $\mathcal{A}^{\mathrm{item}}$ denotes set of all item-level $(\varepsilon, \delta)$-DP algorithms. This directly follows from Kamath et al. [38, Lemma 6.7], concluding the proof.

### D.5 Mean Estimation of Sub-Gaussian Distribution

In this section, we prove error guarantees for mean estimation of sub-Gaussian distributions. We note that known results in mean estimation of Gaussian distributions and moment bounded distributions [38, 39] imply this bound. We include it here for the sake of completeness to demonstrate the strength of our techniques.

**Corollary 5.** *Suppose $P$ is a $\sigma$-sub-Gaussian distribution supported on $[-B, B]^d$ with mean $\mu$. Assume $n \geq (c_1 \sqrt{d \log(1/\delta)}/\varepsilon) \log(B(dn + n^2 \varepsilon^2)/\sigma)$ for a numerical constant $c_1 < \infty$, if $X^n \overset{\mathrm{iid}}{\sim} P$, the output $\mathsf{A}(X^n)$ of Algorithm 2 statisfies*[10]

$$\mathbb{E}\left[\|\mathsf{A}(X^n) - \mu\|_2^2\right] = \tilde{O}\left(\frac{d \sigma^2}{n} + \frac{d^2 \sigma^2}{n^2 \varepsilon^2}\right).$$

*Furthermore, the bound is tight up to logarithmic factors.*

The proof is almost parallel to the proof of Corollary 1 by noting that $X^n$ is $(\sigma \sqrt{d \log(2n/\gamma)}, \gamma)$-concentrated and

$$\mathrm{Var}\left(\frac{1}{n} \sum_{i=1}^{n} X_i\right) = \frac{d \sigma^2}{n}.$$

The tightness of the result follows from Theorem 3.1 and Lemma 3.1 in [18], which proves lower bounds for mean estimation of $k$-dimensional random variables supported on $[-\sigma, \sigma]^k$ under $(\varepsilon, \delta)$-DP constraints.

### D.6 Uniform concentration: answering many queries privately

The statistical query framework subsumes many learning algorithms. For example, we easily express stochastic gradient methods for solving ERM in the language of SQ algorithms (see beginning of Section 4). In the next theorem, we show that with a uniform concentration assumption we can answer a sequence of adaptively chosen queries with variance—or, equivalently, privacy cost—proportional to the concentration radius of the queries instead of the full range.

**Theorem 10.** *If $(Z^n, \mathcal{Q}_B^d)$ is $(\tau, \gamma)$-uniformly concentrated, then for any sequence of (possibly adaptively chosen) queries $\phi_1, \phi_2, \ldots, \phi_K \in \mathcal{Q}_B^d$, there exists an $(\varepsilon, \delta)$-DP algorithm $\mathsf{A}$, such that $\mathsf{A}$ outputs $v_1, v_2, \ldots, v_K$ satisfying $(v_1, v_2, \ldots, v_K) \sim_\beta (v_1', v_2', \ldots, v_K')$, where*

$$\mathbb{E}\left[v_k' | Z^n\right] = \frac{1}{n} \sum_{i=1}^{n} \phi_k(Z_i) \quad \text{and} \quad \mathrm{Var}(v_k' | Z^n) \leq \frac{8 c_0 dK \tau^2 \log(Kdn/\gamma) \log^2(2K/\delta)}{n^2 \varepsilon^2} = \tilde{O}\left(\frac{dK \tau^2}{n^2 \varepsilon^2}\right),$$

---

[10] For precise log factors, see Appendix D.5.

*where $c_0 = 102400$ and $\beta = \min\left\{1, 2\gamma + \frac{d^2 K B \sqrt{\log(dKn/\gamma)}}{\tau} \exp\left(-\frac{n\varepsilon}{48\sqrt{2dK\log(2/\delta)\log(2K/\delta)}}\right)\right\}.$*

The algorithm for Theorem 10 is simply applying Algorithm 2 to $\{\phi_k(Z_i)\}_{i\in[n]}$ with $\varepsilon_0 = \frac{\varepsilon}{2\sqrt{2K\log(2/\delta)}}$ and $\delta_0 = \frac{\delta}{2K}$ for each query. Algorithm 3 is an illustration of an application of this result.

*Proof.* For each query $\phi_k, k \in [K]$, the algorithm computes $\phi_k(Z_i), i \in [n]$ and returns

$$v_k = \textbf{WinsorizedMeanHighD}\big(\{\phi_k(Z_i)\}_{i\in[n]}, \varepsilon_0, \delta_0, \tau, B, \gamma/K\big)$$

where

$$\varepsilon_0 = \frac{\varepsilon}{2\sqrt{2K\log(2/\delta)}}, \quad \delta_0 = \frac{\delta}{2K}.$$

**Privacy guarantee.** The proof is immediate and hinges on the strong-composition theorem. Under the standard strong composition results of [23, Theorem III.3], for any $\delta' \in (0,1]$, the output of Algorithm 3 is $(\bar{\varepsilon}, \bar{\delta})$-user-level DP with

$$\bar{\varepsilon} = K\varepsilon_0(\exp(\varepsilon_0) - 1) + \sqrt{2K\ln(1/\delta')}\varepsilon_0, \qquad \delta = K\delta_0 + \delta'.$$

Plugging in values of $\varepsilon_0, \delta_0$ concludes the proof.

**Utility guarantee.** The proof follows is very similar to the proof of Theorem 2 with $\alpha = \gamma/K$. We conclude by using the subadditivity of the TV distances (or equivalently, a union bound) over all $K$ queries. $\qquad\square$

# E Proofs from Section 4

## E.1 Uniform Concentration

**Proposition 1** (Concentration of random gradients)**.** *Let $S_u \overset{\text{iid}}{\sim} P_u, |S_u| = m$ for $u \in [n]$ and $\alpha \geq 0$. Under Assumptions A3 and A4, with probability greater than $1 - \alpha$ it holds that*

$$\max_{u\in[n]} \sup_{\theta\in\Theta}\|\nabla\mathcal{L}(\theta; S_u) - \nabla\mathcal{L}(\theta; P_u)\|_2 = O\left(\sigma\sqrt{\frac{d\log\left(\frac{RHm}{d\sigma}\right)}{m} + \frac{\log\left(\frac{n}{\alpha}\right)}{m}}\right).$$

*Proof.* The proof relies on a standard covering number argument. We know that $\sup_{\theta_1,\theta_2\in\Theta}\|\theta_1 - \theta_2\| \leq R$. This implies that $\Theta \subset \mathbb{B}_2^d(\theta_0, R)$, where $\mathbb{B}_2^d(v, r)$ is the $d$-dimensional $\ell_2$-ball centered at $v \in \mathbb{R}^d$ of radius $r$. Without loss of generality, we assume $\theta_0 = 0$, i.e. the constraint set $\Theta$ is centered at $0$.

Let us consider $\Gamma_{\|\cdot\|_2}(\Theta, \Delta) =: \Gamma$, a $\Delta$-net of $\Theta$ for the $\ell_2$ norm, i.e. such that $|\Gamma| < \infty$ and that for all $\theta, \vartheta \in \Theta, \|\theta - \vartheta\|_2 \leq \Delta$. Standard results (e.g. Vershynin [56, Corollary 4.2.13]) guarantee that there exists such a set and that its cardinality is smaller than $(1 + 2R/\Delta)^d$.

Since $\ell$ is uniformly $H$-smooth, for any sample $S$ we immediately have that

$$\sup_{\theta\in\Theta}\|\nabla\mathcal{L}(\theta; S) - \nabla\mathcal{L}(\theta; P)\|_2 \leq \max_{\vartheta\in\Gamma}\|\nabla\mathcal{L}(\vartheta; S) - \nabla\mathcal{L}(\vartheta; P)\|_2 + 2H\Delta.$$

Consequently, letting $t > 0$, we have that

$$\mathbb{P}\left(\sup_{\theta\in\Theta}\|\nabla\mathcal{L}(\theta; S) - \nabla\mathcal{L}(\theta; P)\|_2 \geq t\right) \leq \mathbb{P}\left(\max_{\vartheta\in\Gamma}\|\nabla\mathcal{L}(\vartheta; S) - \nabla\mathcal{L}(\vartheta; P)\| \geq t/2\right) + \mathbb{P}(H\Delta \geq t/4).$$

For the second term, we simply need to ensures that when choosing $t$ and $\Delta$, it holds that $H\Delta < t/4$. Let us now bound the first term. Once again, let us consider $\Xi$ a $1/2$-net of $\mathbb{B}_2^d(0, 1)$. For any $v \in \mathbb{R}^d$, it holds that

$$\|v\|_2 = \sup_{\|u\|_2\leq 1}\langle u, v\rangle \leq \max_{\tilde{u}\in\Xi}\langle \tilde{u}, v\rangle + \sup_{w\in\mathbb{B}_2^d(0,1/2)}\langle w, v\rangle = \max_{\tilde{u}\in\Xi}\langle \tilde{u}, v\rangle + \frac{1}{2}\|v\|_2,$$

which implies that $\|v\|_2 \leq 2 \max_{\tilde{u} \in \Xi} \langle \tilde{u}, v \rangle$. Thus,

$$\mathbb{P}\left(\max_{\vartheta \in \Gamma} \|\nabla \mathcal{L}(\vartheta; S) - \nabla \mathcal{L}(\vartheta; P)\|_2 \geq t/2\right) \leq \mathbb{P}\left(\max_{\vartheta \in \Gamma, v \in \Xi} \langle v, \nabla \mathcal{L}(\vartheta; S) - \nabla \mathcal{L}(\vartheta; P) \rangle \geq t/4\right)$$

$$\leq |\Gamma| \cdot |\Xi| e^{-\frac{mt^2}{2\sigma^2}}$$

$$= 5^d \left(1 + \frac{2R}{\Delta}\right)^d e^{-\frac{mt^2}{2\sigma^2}},$$

where the penultimate line follows from a union bound and Assumption A4 which guarantees that $\nabla \mathcal{L}(\vartheta; S)$ is a $\sigma^2/m$-sub-Gaussian vector. We set $t = \sigma \sqrt{\frac{2}{m}(d \log(5 + 10R/\Delta) + \log(n/\alpha))}$. Picking $\Delta = \min\{1, \frac{\sqrt{2}\sigma}{4H}\sqrt{\frac{d}{m}}\}$ and applying a union bound over $n$ points conclude the proof.

$\square$

## E.2 Stochastic gradient algorithms

---

**Algorithm 7** Generic optimization algorithm

---

1: **Input:** Number of steps $T$, stochastic first-order oracle $\mathsf{O}_{F,\nu^2}$, optimization algorithm with $\{\mathcal{O}, \mathsf{Query}, \mathsf{Update}, \mathsf{Aggregate}\}$, initial output $o_0$.
2: **for** $t = 0, \ldots, T - 1$ **do**
3:   $\theta_t \leftarrow \mathsf{Query}(o_t)$.
4:   $g_t \leftarrow \mathsf{O}_{F,\nu^2}(\theta_t)$.
5:   $o_{t+1} \leftarrow \mathsf{Update}(o_t, g_t)$.
6: **end for**
7: **return** $\widehat{\theta}_T \leftarrow \mathsf{Aggregate}(o_0, \ldots, o_T)$.

---

**Proposition 4** (Convergence of stochastic gradient methods). *Let $F : \Theta \to \mathbb{R}$ be an $H$-smooth function. Assume that we have access to a stochastic first-order gradient oracle with variance bounded by $\nu^2$, denoted by $\mathsf{O}_{F,\nu^2}$. In each of the following cases, let $T$ be the desired number of calls to $\mathsf{O}_{F,\nu^2}$, there exist an optimization algorithm—defined by Update, Query and Aggregate and used as in Algorithm 7—with output $\widehat{\theta}_T \in \Theta$ such that the following convergence guarantees hold.*

*(i) [16, Theorem 6.3] Assume $F$ is convex, then it holds that*

$$\mathbb{E}[F(\widehat{\theta}_T) - \inf_{\theta' \in \Theta} F(\theta')] \leq O\left(\frac{HR^2}{T} + \frac{\nu R}{\sqrt{T}}\right). \tag{21}$$

*(ii) [43, Corollary 32] Assume that $F$ is $\mu$-strongly-convex, and that we have access to $\theta_0 \in \Theta$ such that $F(\theta_0) - \inf_{\theta' \in \Theta} F(\theta') \leq \Delta_0$, then it holds that*

$$\mathbb{E}[F(\widehat{\theta}_T) - \inf_{\theta' \in \Theta} F(\theta')] \leq O\left(\Delta_0 \exp\left(-\frac{\mu}{H}T\right) + \frac{\nu^2}{\mu T}\right). \tag{22}$$

*(iii) [20, Corollary 3.6] Let us define the* gradient mapping $\mathsf{G}_{F,\gamma}$

$$\mathsf{G}_{F,\gamma}(\theta) := \frac{1}{\gamma}[\theta - \Pi_\Theta(\theta - \gamma \nabla F(\theta))].$$

*Assume that we have access to $\theta_0$ such that $\|\mathsf{G}_{F,1/H}(\theta_0)\|_2 - \inf_{\theta'} \|\mathsf{G}_{F,1/H}(\theta')\|_2 \leq \Delta_1$, it holds that*

$$\mathbb{E}\|\mathsf{G}_{F,1/H}(\widehat{\theta}_T)\|_2^2 \leq O\left(\frac{H\Delta}{T} + \nu \sqrt{\frac{H\Delta_1}{T}}\right). \tag{23}$$

**Remark 2.** For convex functions, the algorithm is fixed-stepsize, averaged, projected SGD. For strongly-convex functions, the algorithm consists of projected SGD with a fixed stepsize and non-uniform averaging followed by a single restart with decreasing stepsize. Finally, in the non-convex case, the Query and Update sub-routine are also projected SGD with fixed stepsize while the Aggregate selects one of the past iterates uniformly at random.

### E.3 Proof of Theorem 3

**Theorem 3** (Privacy and utility guarantees for ERM). *Assume A2 holds and recall that $\widetilde{G} = \sigma\sqrt{d}$, assume[11] $n = \tilde{\Omega}(\sqrt{dT}/\varepsilon)$ and let $\widehat{\theta}$ be the output of Algorithm 3. There exists variants of projected SGD (e.g. the ones we present in Proposition 4) such that, with probability greater than $1 - \gamma$:*

*(i) If for all $z \in \mathcal{Z}, \ell(\cdot; z)$ is convex, then*

$$\mathbb{E}\left[\mathcal{L}(\widehat{\theta}; \mathcal{S}) - \inf_{\theta' \in \Theta} \mathcal{L}(\theta'; \mathcal{S}) \,\Big|\, \mathcal{S}\right] = \tilde{O}\left(\frac{R^2 H}{T} + R\widetilde{G}\frac{\sqrt{d}}{n\sqrt{m}\varepsilon}\right).$$

*(ii) If for all $z \in \mathcal{Z}, \ell(\cdot; z)$ is $\mu$-strongly-convex, then*

$$\mathbb{E}\left[\mathcal{L}(\widehat{\theta}; \mathcal{S}) - \inf_{\theta' \in \Theta} \mathcal{L}(\theta'; \mathcal{S}) \,\Big|\, \mathcal{S}\right] = \tilde{O}\left(GR \exp\left(-\frac{\mu}{H}T\right) + \widetilde{G}^2 \frac{d}{\mu n^2 m \varepsilon^2}\right).$$

*(iii) Otherwise, defining the* gradient mapping[12] $\mathsf{G}_{F,\gamma}(\theta) := \frac{1}{\gamma}[\theta - \Pi_\Theta(\theta - \gamma \nabla F(\theta))]$, *we have*

$$\mathbb{E}\left[\|\mathsf{G}_{\mathcal{L}(\cdot;\mathcal{S}),1/H}(\widehat{\theta})\|_2^2 | \mathcal{S}\right] = \tilde{O}\left(\frac{H^2 R}{T} + HR\widetilde{G}\frac{\sqrt{d}}{n\sqrt{m}\varepsilon}\right).$$

*For $\varepsilon \leq 1, \delta > 0$, Algorithm 3 instantiated with any first-order gradient algorithm is $(\varepsilon, \delta)$-user-level DP. In the case that only A1 holds, the same guarantees hold whenever $\Delta \leq \mathsf{poly}(d, \frac{1}{n}, \frac{1}{m}, \frac{1}{\varepsilon})$.*

*Proof.* First note that the gradient estimation steps (Step 5 and 6) in Algorithm 3 can be viewed as answering $T$ adaptively chosen queries.

**Privacy guarantees.** The privacy guarantee follows directly from Theorem 10.

**Utility guarantees.** By Proposition 1, we have the gradients are $(\tau, \gamma/3)$-concentrated with $\tau = \sigma\sqrt{d\log\left(\frac{RHm}{d\sigma}\right)/m + \log\left(\frac{3n}{\gamma}\right)/m}$. Hence, Theorem 10 guarantees that

$$(\bar{g}_0, \ldots, \bar{g}_{T-1}) \sim_\beta (\bar{g}'_0, \ldots, \bar{g}'_{T-1}),$$

where $\beta = \min\left\{1, \frac{2\gamma}{3} + \frac{d^2 TB\sqrt{\log(3dTn/\gamma)}}{\tau}\exp\left(-\frac{n\varepsilon}{48\sqrt{2dT\log(2/\delta)\log(2T/\delta)}}\right)\right\}$ and $\forall i \in [T], \bar{g}'_0$ is from $\mathsf{O}_{\mathcal{L}(\cdot;\mathcal{S}),\nu^2}(\theta_t)$ with

$$\nu^2 \leq \frac{8c_0 dT\tau^2 \log(3Tdn/\gamma)\log^2(2T/\delta)}{n^2\varepsilon^2} \leq \frac{8c_0 d^2 T\sigma^2 \log(3Tdn/\gamma)\log^2(2T/\delta)\log(3RHmn/d\sigma\gamma)}{n^2\varepsilon^2}.$$

Moreover, when $n \geq \tilde{\Omega}(1)\sqrt{dT\log(2/\delta)\log(2T/\delta)\log(dmTB/\sigma\gamma)}/\varepsilon$, where $\tilde{\Omega}(1)$ hides log-log factors, we have $\beta < \gamma$.

**Convergence rates** Finally, depending on the assumptions on the function $\mathcal{L}(\cdot; \mathcal{S})$, we use the various results of Proposition 4 for the value of $\nu$ above. To make the results simpler we note that for (ii) of Proposition 4, we upper bound $\Delta_0$ by $GR$ and for (iii), we upper bound $\Delta_1$ by $HR^2$. This concludes the proof.

$\square$

## F   Proofs for Section 5

### F.1   Proofs for Theorem 4

We begin with a result that guarantees that the (regularized) empirical risk minimizer has good generalization properties. It relies on a combination of convex analysis and stability arguments. This proof exists in the literature (see, e.g. [51]), we add it here for completeness and with some small variation: (1) that the optimization is constrained (2) that Assumption A4 might improve stability when $\sigma\sqrt{d} \leq G$.

---

[11]For precise log factors, see Appendix E.3.

[12]In the unconstrained case—$\Theta = \mathbb{R}^d$—this corresponds to an $\epsilon$-stationary point as $\mathsf{G}_{F,\gamma}(x) = \nabla F(x)$.

**Proposition 5** (Generalization properties of regularized ERM). *Let* $(Z_1, \ldots, Z_N) \overset{\text{iid}}{\sim} P$. *Let* $\ell : \Theta \times \mathcal{Z} \to \mathbb{R}$ *be convex, G-Lipschitz with respect to the* $\|\cdot\|_2$ *and such that Assumption A4 holds. Let us denote* $\underline{G} = \min\{G, \sigma\sqrt{d}\}$. *Let*

$$\theta_{S,\lambda,\vartheta}^* := \underset{\theta \in \Theta}{\operatorname{argmin}}\left\{\mathcal{L}(\theta; S) + \frac{\lambda}{2}\|\theta - \vartheta\|_2^2\right\}.$$

*The following holds*

$$\mathbb{E}\big[\mathcal{L}(\theta_{S,\lambda,\vartheta}^*; P)\big] - \mathcal{L}(\theta; P) \le \frac{\lambda}{2}\mathbb{E}\big[\|\theta - \vartheta\|_2^2\big] + \tilde{O}(1)\frac{G\underline{G}}{N\lambda}, \;\; \text{for all } \; \theta \in \Theta. \qquad (24)$$

*Proof.* We first show the stability of the minimizer of the regularized empirical risk. Let us consider $S_0 = \{Z_1, \ldots, Z_N\}$ and $S_1 = \{Z_1', \ldots, Z_N'\}$ where $Z_j = Z_j'$ for all $j \neq i$ in $[N]$. We first show that

$$\big\|\theta_{S,\lambda,\vartheta}^* - \theta_{S',\lambda,\vartheta}^*\big\|_2 \le \tilde{O}(1)\frac{\underline{G}}{N\lambda}.$$

For conciseness, we denote $\mathcal{L}_b(\theta) := \mathcal{L}(\theta; S_b) + \frac{\lambda}{2}\|\theta - \vartheta\|_2^2$ and $\theta_{S_b,\lambda,\vartheta}^* = \theta_b$ for $b \in \{0, 1\}$. Since $\mathcal{L}_0$ is $\lambda$-strongly-convex, its gradients are co-coercive, meaning

$$\frac{\lambda}{2}\|\theta_0 - \theta_1\|_2^2 \le \langle \nabla\mathcal{L}_0(\theta_0) - \nabla\mathcal{L}_0(\theta_1), \theta_0 - \theta_1\rangle.$$

First, let us note that $\nabla\mathcal{L}_0(\theta_1) = \nabla\mathcal{L}_1(\theta_1) + \frac{1}{N}(\nabla\ell(\theta_1; Z_i) - \nabla\ell(\theta_1; Z_i'))$. In other words,

$$\frac{\lambda}{2}\|\theta_0 - \theta_1\|_2^2 \le \langle\nabla\mathcal{L}_0(\theta_0), \theta_0 - \theta_1\rangle + \langle\nabla\mathcal{L}_1(\theta_1), \theta_1 - \theta_0\rangle + \frac{1}{N}\langle\nabla\ell(\theta_1; Z_i) - \nabla\ell(\theta_1; Z_i'), \theta_1 - \theta_0\rangle.$$

Since $\theta_b$ is the minimizer of $\mathcal{L}_b(\cdot)$ constrained in $\Theta$ for $b \in \{0, 1\}$, by first-order optimiality condition, it holds that

$$\langle\nabla\mathcal{L}_b(\theta_b), \theta_b - \theta_{1-b}\rangle \le 0.$$

Consequently,

$$\frac{\lambda}{2}\|\theta_0 - \theta_1\|_2^2 \le \frac{1}{N}\langle\nabla\ell(\theta_1; Z_i) - \nabla\ell(\theta_1; Z_i'), \theta_1 - \theta_0\rangle \le \frac{1}{N}\|\nabla\ell(\theta_1; Z_i) - \nabla\ell(\theta_1; Z_i')\|_2\|\theta_1 - \theta_0\|_2.$$

Since $\ell(\cdot; z)$ is $G$-Lipschitz for all $z \in \mathcal{Z}$, we have that $\|\nabla\ell(\theta_1; Z_i) - \nabla\ell(\theta_1; Z_i')\|_2 \le 2G$. However, with the addition of Assumption A4, Proposition 1 (applied with $m = 1$) guarantees that with probability greater than $1 - \alpha$,

$$\sup_{\theta \in \Theta}\|\nabla\mathcal{L}(\theta; Z_i) - \nabla\mathcal{L}(\theta; P)\| \le \tilde{O}(1)\,\sigma\sqrt{d},$$

where we note that the dependence is only *logarithmic* in $\alpha$. This immediately yields that with probability greater than $1 - \alpha$,

$$\frac{\lambda}{2}\|\theta_0 - \theta_1\|_2 \le \tilde{O}(1)\frac{\underline{G}}{\lambda N}.$$

Finally, this implies that

$$\text{for all } z \in \mathcal{Z}, \mathbb{E}[|\ell(\theta_0; z) - \ell(\theta_1; z)|] \le G\mathbb{E}[\|\theta_0 - \theta_1\|_2] \le \tilde{O}(1)\frac{G\underline{G}}{\lambda N},$$

by $G$-Lipschitzness of $\ell$ and setting $\alpha = \frac{\underline{G}}{\lambda N R}$, or in the language of stability (see e.g. [14]), $S \to \theta_{S,\lambda,\vartheta}^*$ is $\frac{G\underline{G}}{\lambda N}$-uniformly-stable. Standard stability arguments let us conclude the proof. $\qquad\square$

We now state and prove Theorem 4.

**Theorem 4** (Phased ERM for SCO). *Algorithm 4 is user-level* $(\varepsilon, \delta)$-*DP. When A2 holds and* $n = \tilde{\Omega}(\min\{\sqrt[3]{d^2mH^2R^2/(G\underline{G}\varepsilon^4)}, HR\sqrt{m}/(\sigma\varepsilon)\})$, *or, equivalently,* $H = \tilde{O}(\sqrt{\frac{n^2\varepsilon^2\sigma^2}{R^2m} + \frac{G\underline{G}n^3\varepsilon^4}{d^2R^2m}})$ *for all* $P$ *and* $\ell$ *satisfying Assumptions A3 and A4, we have*

$$\mathbb{E}\left[\mathcal{L}(\mathsf{A}_{\mathsf{PhasedERM}}(\mathcal{S}); P_0)\right] - \min_{\theta' \in \Theta}\mathcal{L}(\theta'; P_0) = \tilde{O}\left(\frac{R\sqrt{G\underline{G}}}{\sqrt{mn}} + R\widetilde{G}\frac{\sqrt{d}}{n\sqrt{m}\varepsilon}\right).$$

*Furthermore, our results still hold in the heterogeneous setting (Assumption A1) whenever* $\Delta \le \mathsf{poly}(d, \frac{1}{n}, \frac{1}{m}, \frac{1}{\varepsilon})$; *the risk guarantee being with respect to any user distribution* $P_u$.

*Proof.* The proof hinges on repeatedly using of Corollary 2 and Proposition 5 after decomposing the excess risk. Recall that $\widehat{\theta}_t$ is the output of round $t$ i.e.

$$\widehat{\theta}_t \approx \underset{\theta \in \Theta}{\operatorname{argmin}} \, \mathcal{L}(\theta; S_t) + \frac{\lambda_t}{2} \|\theta - \widehat{\theta}_{t-1}\|_2^2.$$

We denote by $\theta_t^*$ the *true* minimizer at round $t$ i.e.

$$\theta_t^* := \underset{\theta \in \Theta}{\operatorname{argmin}} \, \mathcal{L}(\theta; S_t) + \frac{\lambda_t}{2} \|\theta - \widehat{\theta}_{t-1}\|_2^2.$$

Let us denote $\theta^* = \operatorname{argmin}_{\theta \in \Theta} \mathcal{L}(\theta; P)$, we decompose the regret in the following way

$$\mathbb{E}[\mathcal{L}(\widehat{\theta}_T; P) - \mathcal{L}(\theta^*; P)] = \underbrace{\mathbb{E}\Big[\mathcal{L}(\widehat{\theta}_T; P) - \mathcal{L}(\theta_T^*; P)\Big]}_{=:\Delta_0} + \underbrace{\sum_{t=2}^{T} \mathbb{E}\big[\mathcal{L}(\theta_t^*; P) - \mathcal{L}(\theta_{t-1}^*; P)\big]}_{=:\Delta_1}$$
$$+ \underbrace{\mathbb{E}[\mathcal{L}(\theta_1^*; P) - \mathcal{L}(\theta^*; P)]}_{=:\Delta_2}.$$

By Proposition 5 and because $\Theta$ is bounded by $R$, we directly have that

$$\Delta_2 \leq \frac{\lambda_1 R^2}{2} + \tilde{O}\left(\frac{G\underline{G}}{\lambda_1 n_1 m}\right).$$

Turning to $\Delta_1$, for every $t \in \{2, \ldots, T\}$, again by Proposition 5, it holds that

$$\mathbb{E}\big[\mathcal{L}(\theta_t^*; P) - \mathcal{L}(\theta_{t-1}^*; P)\big] \leq \frac{\lambda_t}{2} \mathbb{E}\Big[\|\theta_{t-1}^* - \widehat{\theta}_{t-1}\|_2^2\Big] + \tilde{O}\left(\frac{G\underline{G}}{\lambda_t n_t m}\right).$$
$$\leq \tilde{O}\left(\frac{\lambda_t}{2} \frac{\sigma^2 d^2}{\lambda_{t-1}^2 n_{t-1}^2 m \varepsilon^2} + \frac{G\underline{G}}{\lambda_t n_t m}\right)$$
$$\leq \tilde{O}\left(\frac{\sigma^2 d^2}{\lambda_{t-1} n_{t-1}^2 m \varepsilon^2} + \frac{G\underline{G}}{\lambda_t n_t m}\right),$$

where the second inequality is an application of Corollary 2[13] and the third is because $\lambda_{t-1} = \lambda_t/4$. Noting that $\lambda_{t-1} n_{t-1}^2 = 2^{t-1} \lambda n$, we have

$$\Delta_1 \leq \tilde{O}\left((T-1)\frac{\sigma^2 d^2}{\lambda n^2 m \varepsilon^2} + \frac{G\underline{G}}{\lambda n m} \sum_{t=2}^{T} 2^{-t}\right) = \tilde{O}\left(\frac{\sigma^2 d^2}{\lambda n^2 m \varepsilon^2} + \frac{G\underline{G}}{\lambda n m}\right),$$

where we use that $T$ is logarithmic. Finally, using Corollary 2, and that $\mathcal{L}(\cdot; P)$ is $G$-Lipschitz, we have that

$$\Delta_0 \leq \mathbb{E}\Big[G\|\theta_T^* - \widehat{\theta}_T\|_2\Big] \leq G\sqrt{\mathbb{E}\Big[\|\theta_T^* - \widehat{\theta}_T\|_2^2\Big]} = \tilde{O}\left(\frac{G\sigma d}{2^T \lambda n \sqrt{m} \varepsilon}\right).$$

Combining the upper bounds, we have

$$\mathbb{E}\Big[\mathcal{L}(\widehat{\theta}_T; P) - \mathcal{L}(\theta^*; P)\Big] = \tilde{O}\left(\frac{G\sigma d}{2^T \lambda n \sqrt{m} \varepsilon} + \frac{\sigma^2 d^2}{\lambda n^2 m \varepsilon^2} + \frac{G\underline{G}}{\lambda n m} + \frac{\lambda_1 R^2}{2} + \frac{G\underline{G}}{\lambda_1 n_1 m}\right),$$

and setting $T = \lceil \log_2(\frac{Gn\sqrt{m}\varepsilon}{\sigma d}) \rceil$ and $\lambda = \sqrt{\frac{\sigma^2 d^2}{n^2 m \varepsilon^2} + \frac{G\underline{G}}{nm}}/R$ yields the final result.

$\square$

---

[13]The condition on $n$ for the corollary holds when the condition on $n$ is satisfied in the Theorem statement.

### F.2    Proofs of Theorem 5

**Theorem 5** (Lower bound for SCO). *There exists a distribution $P$ and a loss $\ell$ satisfying Assumptions A3 and A4 such that for any algorithm $\mathsf{A}$ satisfying $(\varepsilon, \delta)$-DP at user-level, we have*

$$\mathbb{E}\left[\mathcal{L}(\mathsf{A}(\mathcal{S}); P)\right] - \min_{\theta' \in \Theta} \mathcal{L}(\theta'; P) = \Omega\left(\frac{RG}{\sqrt{mn}} + RG\frac{\sqrt{d}}{n\sqrt{m}\varepsilon}\right).$$

The first term is a lower bound for SCO without any constraints [44, 2]. We only prove the second term here. Note that without loss of generality, we can assume $G \geq 20\sigma\sqrt{d}$ and prove a lower bound of $\Omega(R\widetilde{G}\sqrt{d}/n\sqrt{m}\varepsilon)$. Else, we set $\sigma' = G/(20\sqrt{d})$ and embed the original problem into a lower-dimensional (thus easier) problem where the gradients are $\sigma'^2$ sub-Gaussian. In the rest of the section, we consider $\Theta = \mathbb{B}_2^d(0, R)$ for $R > 0$. As we explained in Section 5.2, we consider the following loss[14]

$$\ell(\theta; z) := -\langle \theta, z \rangle.$$

Finally, we define (a collection) of data distributions. For a mean $\mu \in \mathbb{R}^d$, a covariance $\Sigma \in \mathbb{R}^{d \times d}$ and $B > 0$, we consider the family of $\ell_\infty$-truncated Gaussians. Recall the definition in (20),

$$Z \sim \mathsf{N}^{\mathrm{tr}}(\mu, \Sigma, B) \text{ if } Z_0 \sim \mathsf{N}(\mu, \Sigma) \text{ and set for all } j \in [d] \ Z(j) = \frac{Z_0(i)}{\max\{1, |Z_0(j)|/B\}}.$$

In other words, the standard high-dimension Gaussian distribution where the mass outside of $\mathbb{B}_\infty^d(0, B)$ has been radially projected back on the sphere on each dimension.

Consequently, considering the data distribution $P = \mathsf{N}^{\mathrm{tr}}(\mu, \sigma^2 I_d, G/\sqrt{d})$, $\ell$ is almost surely $G$-Lipschitz. Additionally, both assumptions A3 and A4 hold.

We now formally state the reduction from SCO to Gaussian mean-estimation. The main difficulty is that the mean of $\mathsf{N}^{\mathrm{tr}}(\mu, \sigma^2 I_d, G/\sqrt{d})$ and $\mathsf{N}(\mu, \sigma^2 I_d)$ do not coincide. However, we show that when $G$ is sufficiently large compared to $\|\mu\|_2$—which implies that we rarely clip—then the reduction holds.

**Proposition 6** (Reduction from SCO to Gaussian mean estimation with item-level DP constraints). *Let $B > 0, \sigma > 0, G > 0$ such that $B + 10\sigma\sqrt{d} < G$, we consider the following collections of distributions*

$$\mathcal{P}_{\sigma, B} := \left\{\mathsf{N}(\mu, \sigma^2 I_d) : \|\mu\|_2 \in [B/2, B]\right\} \text{ and } \mathcal{P}_{\sigma, B, G/\sqrt{d}}^{\mathrm{tr}} := \left\{\mathsf{N}^{\mathrm{tr}}(\mu, \sigma^2 I_d, G/\sqrt{d}) : \|\mu\|_2 \in [B/2, B]\right\}.$$

*The following reduction holds*

$$\inf_{\substack{\mathsf{A}: \mathcal{Z} \to \Theta \\ \mathsf{A} \in \mathcal{A}_{\varepsilon, \delta}^{\mathrm{item}}}} \sup_{P \in \mathcal{P}_{\sigma, B, G/\sqrt{d}}^{\mathrm{tr}}} \mathbb{E}_P\left[\mathcal{L}(\mathsf{A}(Z^n); P) - \inf_{\theta' \in \Theta} \mathcal{L}(\theta'; P)\right] \geq$$

$$\frac{BR}{4} \inf_{\substack{\widehat{u}: \mathcal{Z} \to \mathbb{S}^{d-1} \\ \widehat{u} \in \mathcal{A}_{\varepsilon, \delta}^{\mathrm{item}}}} \sup_{P \in \mathcal{P}_{\sigma, B}} \mathbb{E}_P\left[\|\widehat{u}(Z^n) - \mu/\|\mu\|_2\|_2^2\right] + O\left(R\sigma e^{-10d}\right),$$

*where we recall that $\mathcal{A}_{\varepsilon, \delta}^{\mathrm{item}}$ is the set of $(\varepsilon, \delta)$-item-level DP algorithm for which the domain and co-domain are clear from context.*

Before proving the proposition, we prove a Lemma that says, as previewed, that when $G$ is large enough compared to $\|\mu\|_2$ and $\sigma$, then the expectation of $\mathsf{N}^{\mathrm{tr}}(\mu, \sigma^2 I_d, G/\sqrt{d})$ and $\mu$ are exponentially close in $\ell_2$-norm.

**Lemma 3.** *Suppose $\|\mu\|_2 + 10\sqrt{d}\sigma < G$,*

$$\|\mathbb{E}_{Z \sim \mathsf{N}^{\mathrm{tr}}(\mu, \sigma^2 I_d, G/\sqrt{d})}[Z] - \mu\|_2 = O\left(\sigma e^{-10d}\right).$$

---

[14] The negative sign is here for convenience; a positive sign would entail reducing it to finding the *negative* normalized mean.

*Proof of Lemma 3.* It would be enough to show that $\forall i \in [d]$,

$$|\mathbb{E}_{Z \sim \mathsf{N}^{\mathrm{tr}}(\mu, \sigma^2 I_d, G/\sqrt{d})} [Z](i) - \mu(i)| = O(\sigma e^{-10d}/\sqrt{d}).$$

Let $\alpha = \frac{\mu(i) + G/\sqrt{d}}{\sigma}$, $\beta = \frac{\mu(i) - G/\sqrt{d}}{\sigma}$ and $\phi(x) = \frac{1}{\sqrt{2\pi}} e^{-\frac{1}{2}x^2}$ be the density function of $N(0,1)$. We have

$$\mathbb{E}_{Z \sim \mathsf{N}^{\mathrm{tr}}(\mu, \sigma^2 I_d, G/\sqrt{d})} [Z](i) = \mu(i) - \sigma \frac{\phi(\alpha) - \phi(\beta)}{\int_\alpha^\beta \phi(x) dx}.$$

Plugging in $\|\mu\|_2 + 10\sqrt{d}\sigma < G$ we obtain the lemma. $\square$

We can now prove the proposition.

*Proof.* Let $P = \mathsf{N}^{\mathrm{tr}}(\mu, \sigma^2 I_d, G/\sqrt{d})$ and denote $\mu^{\mathrm{tr}} = \mathbb{E}_P[Z]$ the mean of the truncated Gaussian. We consider $\theta_0 = -R\frac{\mu}{\|\mu\|_2}$, in other words the minimum of $\mathcal{L}(\theta; P)$, if the Gaussian was not truncated. Let $\theta \in \Theta$, we have that

$$
\begin{aligned}
\mathcal{L}(\theta; P) - \mathcal{L}(\theta^*; P) &\geq -\mathcal{L}(\theta; P) - \mathcal{L}(\theta_0; P) \\
&= -\langle \theta - \theta_0, \mu^{\mathrm{tr}} \rangle \\
&= -\langle \theta - \theta_0, \mu \rangle - \langle \theta - \theta_0, \mu^{\mathrm{tr}} - \mu \rangle \\
&\geq -\langle \theta - \theta_0, \mu \rangle + 2 \inf_{\theta'} \langle \theta', \mu^{\mathrm{tr}} - \mu \rangle \\
&= -\langle \theta - \theta_0, \mu \rangle + O(R\sigma e^{-10d}),
\end{aligned}
$$

where the final line uses the fact that $\inf_{\|v\|_2 \leq R} \langle u, v \rangle = -R\|u\|_2$ and Lemma 3.

Moreover, we have

$$
\begin{aligned}
\langle \theta_0 - \theta, \mu \rangle &= R\|\mu\|_2 \left( 1 - \left\langle \frac{\theta}{R}, \frac{\mu}{\|\mu\|_2} \right\rangle \right) \\
&\geq \frac{R\|\mu\|_2}{2} \left( \left\| \frac{\theta}{R} \right\|_2^2 + \left\| \frac{\mu}{\|\mu\|_2} \right\|_2^2 - 2\left\langle \frac{\theta}{R}, \frac{\mu}{\|\mu\|_2} \right\rangle \right) \\
&= \frac{R\|\mu\|_2}{2} \left\| \frac{\theta}{R} - \frac{\mu}{\|\mu\|_2} \right\|_2^2,
\end{aligned}
$$

where we used that $\|\theta/R\| \leq 1$ and completed the square.

We now finally prove the main statement of the proposition. The first observation is that, since the loss is linear, we only need to consider estimators $\mathsf{A} : \mathcal{Z}^n \to \Theta$ such that $\|\mathsf{A}(z^n)\|_2 = R$ for all

$z^n \in \mathcal{Z}^n$, as the minimum is always on the boundary[15]. Consequently, we have

$$\inf_{\mathsf{A}:|\mathsf{A}|_2 \leq R} \sup_{P \in \mathcal{P}^{\mathrm{tr}}_{\sigma,B,G/\sqrt{d}}} \mathbb{E}[\mathcal{L}(\mathsf{A}(Z^n); P) - \min_{\theta' \in \Theta} \mathcal{L}(\theta'; P)]$$

$$= \inf_{\mathsf{A}:|\mathsf{A}|_2 = R} \sup_{P \in \mathcal{P}^{\mathrm{tr}}_{\sigma,B,G/\sqrt{d}}} \mathbb{E}[\mathcal{L}(\mathsf{A}(Z^n); P) - \min_{\theta' \in \Theta} \mathcal{L}(\theta'; P)]$$

$$\geq \inf_{\mathsf{A}:|\mathsf{A}|_2 = R} \sup_{P \in \mathcal{P}^{\mathrm{tr}}_{\sigma,B,G/\sqrt{d}}} \mathbb{E}\frac{R\|\mu\|_2}{2}\left\|\frac{\mathsf{A}(Z^n)}{R} - \frac{\mu}{\|\mu\|_2}\right\|_2^2 + O(R\rho e^{-10d})$$

$$\geq \inf_{\mathsf{A}:|\mathsf{A}|_2 = R} \sup_{P \in \mathcal{P}^{\mathrm{tr}}_{\sigma,B,G/\sqrt{d}}} \frac{RB}{4}\mathbb{E}\left\|\frac{\mathsf{A}(Z^n)}{R} - \frac{\mu}{\|\mu\|_2}\right\|_2^2 + O(R\rho e^{-10d})$$

$$= \inf_{\widehat{u}:\|\hat{u}\|=1} \sup_{P \in \mathcal{P}^{\mathrm{tr}}_{\sigma,B,G/\sqrt{d}}} \frac{RB}{4}\mathbb{E}\left\|\widehat{u}(Z^n) - \frac{\mu}{\|\mu\|_2}\right\|_2^2 + O(R\rho e^{-10d})$$

$$\geq \inf_{\widehat{u}:\|\hat{u}\|=1} \sup_{P \in \mathcal{P}_{\sigma,B}} \frac{RB}{4}\mathbb{E}\left\|\widehat{u}(Z^n) - \frac{\mu}{\|\mu\|_2}\right\|_2^2 + O(R\rho e^{-10d}),$$

where the last line uses that we can always sample from $\mathsf{N}^{\mathrm{tr}}(\mu, \sigma^2 I_d, G/\sqrt{d})$ using samples from $\mathsf{N}(\mu, \sigma^2 I_d)$ and truncating them, thus the problem over $\mathcal{P}^{\mathrm{tr}}_{\sigma,B,G}$ is harder than over $\mathcal{P}_{\sigma,B}$. This concludes the proof. $\qquad\square$

Because of this reduction, for the remainder of this proof we consider Gaussian mean estimation with user-level DP constraints. Recall that in this setting, we have $n$ users, each having $m$ i.i.d. samples from $\mathsf{N}(\mu, \sigma^2 I_d)$. However, the lower bound of [38] only holds for *item-level* DP constraints. In the next proposition, we show that mean estimation of $\mathsf{N}(\mu, \sigma^2 I_d)$ with $n$ users and $m$ samples per user under user-level DP constraints is equivalent to mean estimation of $\mathsf{N}(\mu, \frac{\sigma^2}{m} I_d)$ with $n$ samples under *item-level constraints*. In other words, any user-level DP estimator taking as input $n \cdot m$ samples is equivalent to an item-level DP estimator taking as input $n$ samples corresponding the each user's average.

**Proposition 3** (From multiple samples to one good sample). *Suppose each user $u \in [n]$ observe* $(Z_1^{(u)}, \ldots, Z_m^{(u)}) \overset{\mathrm{iid}}{\sim} \mathsf{N}(\mu, \sigma^2 I_d)$. *For any $(\varepsilon, \delta)$ user-level DP algorithm $\mathsf{A}^{\mathsf{user}}$, there exists an $(\varepsilon, \delta)$-item-level DP algorithm $\mathsf{A}^{\mathsf{item}}$ that takes as input $(\bar{Z}^{(1)}, \ldots, \bar{Z}^{(n)})$ with $\bar{Z}^{(u)} := \frac{1}{m} \sum_{j \leq m} Z_j^{(u)}$ and has the same performance as $\mathsf{A}^{\mathsf{user}}$.*

*Proof.* First of all, note that for Gaussians with unknown mean but known variance, the sample mean is a sufficient statistic. As such, we have that for all $u \in [n]$

$$\text{the distribution of } (Z_1^{(u)}, \ldots, Z_m^{(u)})|\bar{Z}^{(u)} \text{ does not depend on } \mu.$$

Let us now consider an arbitrary user-level DP estimator $\mathsf{A}^{\mathsf{user}}$ and show how to construct an equivalent item-level DP estimator. When provided with $(\bar{Z}^{(1)}, \ldots, \bar{Z}^{(n)})$, for each $j \leq m$, we can sample

$$\tilde{S}_u = (\tilde{Z}_1^{(u)}, \ldots \tilde{Z}_m^{(u)}) \overset{\mathrm{iid}}{\sim} (Z_1^{(u)}, \ldots, Z_m^{(u)})|\bar{Z}^{(u)} \tag{25}$$

and return $\mathsf{A}^{\mathsf{item}}((\bar{Z}^{(u)})_{u \leq n}) = \mathsf{A}^{\mathsf{user}}((\tilde{S}_1, \ldots, \tilde{S}_n))$. Since the distributions are equal given $\bar{Z}^{(u)}$, in expectation the error is the same. $\qquad\square$

This proposition allows us to reduce Gaussian mean estimation with user-level DP, to Gaussian mean estimation with item-level DP albeit with the variance divided by $m$. We thus conclude with (a

---

[15]To make this rigorous, we consider Yao's minimax principe. It holds that $\min_{\mathsf{A}:|\mathsf{A}|_2 \leq R} \max_\mu \mathbb{E}[\bar{\mathcal{L}}(\mathsf{A}(Z^n); P)] = \max_{\mathcal{D}} \min_{\mathsf{A}:|\mathsf{A}|_2 \leq R} \mathbb{E}_{\mu \sim \mathcal{D}} \mathbb{E}[\bar{\mathcal{L}}(\mathsf{A}(Z^n); P)|\mu]$ where $\bar{\mathcal{L}}(\theta; P) := \mathcal{L}(\theta; P) - \inf_{\theta'} \mathcal{L}(\theta'; P)$ and $\mathcal{D}$ is a prior over $\mu$. For a given prior $\mathcal{D}$, the Bayes optimal classifier is the minimum of the posterior mean, which means that $\mathsf{A}(Z^n)$ minimizes $\langle \theta, \mathbb{E}[\mu|Z^n]\rangle$ over $\mathbb{B}_2^d(0, R)$ and thus has norm $R$. We can thus constrain the class of estimators to be of norm exactly $R$ for any prior $\mathcal{D}$. Another application of Yao's minimax principle guarantees that this is also the case for the original (minimax) problem.

slight modification of) the results of [38]. Indeed, we differ only in that their results show that mean estimation is hard, whereas we require that estimating the *direction of the mean* is hard.

First, let us recall the a modified version of the result in [38][16].

**Proposition 7** (Kamath et al. [38, Lemma 6.7]). *Let $Z^n \overset{\text{iid}}{\sim} \mathsf{N}(\mu, \sigma^2 I_d)$ and assume $\delta \leq \frac{\sqrt{d}}{48\sqrt{2}Bn\sqrt{\log(100Rn/\sqrt{d})}}$, then it holds that if $n < d\sigma/(512B\varepsilon)$,*

$$\inf_{\hat{\mu}, \widehat{\mu} \in \mathcal{A}_{\varepsilon,\delta}^{\text{item}}} \sup_{\mu: B/2 \leq \|\mu\|_2 \leq B} \mathbb{E}\big[\|\widehat{\mu}(Z^n) - \mu\|_2^2\big] \geq \frac{B^2}{6}.$$

**Corollary 6** (Estimating the direction of the mean is hard). *Let $Z^n \overset{\text{iid}}{\sim} \mathsf{N}(\mu, \sigma^2 I_d)$, set $B = \frac{d\rho}{512n\varepsilon}$ and assume that $\delta \leq \frac{\sqrt{d}}{48\sqrt{2}Bn\sqrt{\log(100Rn/\sqrt{d})}}$, then it holds that if $n < d\sigma/(512B\varepsilon)$,*

$$\inf_{\substack{\widehat{u}: \|\widehat{u}\|_2 = 1 \\ \widehat{u} \in \mathcal{A}_{\varepsilon,\delta}^{\text{item}}}} \sup_{P \in \mathcal{P}_{\sigma,B}} \mathbb{E}\left[\left\|\widehat{u}(Z^n) - \frac{\mu}{\|\mu\|_2}\right\|_2^2\right] \geq \frac{1}{10}.$$

*Proof.* We prove the corollary by contradiction. Assume there exists an $(\varepsilon, \delta)$-DP estimator $\widehat{u}$ such that

$$\sup_{P \in \mathcal{P}_{\sigma,B}} \mathbb{E}\left[\left\|\widehat{u}(Z^n) - \frac{\mu}{\|\mu\|_2}\right\|_2^2\right] < \frac{1}{10}.$$

Then let $\widehat{\mu} = \frac{3}{4B}\widehat{u}$,

$$
\begin{aligned}
\mathbb{E}[\|\widehat{\mu}(Z^n) - \mu\|_2^2] &= \mathbb{E}\left[\left\|\frac{3B}{4}\widehat{u}(Z^n) - \mu\right\|_2^2\right] \\
&\leq \mathbb{E}\left[\left\|\frac{3B}{4}\widehat{u}(Z^n) - \|\mu\|_2 \cdot \widehat{u}(Z^n)\right\|\right] + \mathbb{E}\left[\|\|\mu\|_2 \cdot \widehat{u}(Z^n) - \mu\|_2^2\right] \\
&= \left(\frac{3B}{4} - \|\mu\|_2\right)^2 + \|\mu\|_2^2\,\mathbb{E}\left[\left\|\widehat{u}(Z^n) - \frac{\mu}{\|\mu\|_2}\right\|_2^2\right] \\
&\leq \frac{B^2}{16} + \frac{B^2}{10} \\
&< \frac{B^2}{6},
\end{aligned}
$$

which contradicts with Proposition 7. $\qquad\square$

Applying Corollary 6 with $B = \sigma/\sqrt{m}$ concludes the proof of the lower bound.

---

[16]It is not guaranteed in the lower bound construction of [38] that $B/2 \leq \|\mu\|_2 \leq B$. In their construction, the mean is taken uniformly from $[-\sqrt{2}B/\sqrt{d}, \sqrt{2}B/\sqrt{d}]^d$. However, the probability that the mean in the lower bound construction being out of this range is exponentially small in $d$. Hence the same lower bound can be obtained by straightforward modifications of the construction.