# OpenReview forum: "Learning with User-Level Privacy"
_NeurIPS.cc/2021/Conference — NeurIPS 2021 Poster_

### Official Review · Reviewer_GqWK · 2021-06-26

**Rating:** 7
**Confidence:** 3

**Summary:**

This paper provides algorithms for various learning tasks under the constraint of user-level differential privacy. Specifically, they provide solutions for problems like mean estimation, empirical risk minimisation, stochastic convex optimisation, and learning hypothesis classes with finite metric entropy. They also provide lower bounds showing optimality of their algorithms for mean estimation and stochastic convex optimisation for different settings.

For mean estimation, they consider 1D distributions (and extend to higher dimensions, too) that are bounded within an interval, but are concentrated in some smaller sub-interval. Using this, they show that for uniformly concentrated queries, they can adaptively answer $K$ queries with a small privacy cost depending polynomially in the concentration parameter ($\tau$) and $K$.

For empirical risk minimisation and stochastic convex optimisation, they assume that each user's data is drawn i.i.d. from different distribution, such that all the distributions are close in TV distance.

They also show that even when each user contributes infinite samples, there would still be some error.

**Limitations And Societal Impact:**

Yes, they have. But they don't seem too negative in the sense that there are limits of differential privacy in terms of practice right now. They may be overcome in future.

**Main Review:**

Update: On the authors' response, I have decided to update my final evaluation.

Strengths:
1. The paper studies these problems under a more stringent notion of user-level differential privacy. This is not just building up on prior work incrementally.
2. The math involved is fairly rigorous, and the paper is technical enough for this conference.
3. The methods and frameworks provided in Theorems 2 and 3 are fairly simple, hence practical.
4. The paper feels fairly well-written to me, with all the preliminaries provided properly.
5. Algorithm 4 looks quite neat. It is simple and efficient, and uses their own algorithm for empirical risk minimisation.
6. Their results are optimal, at least for two types of problems under distributional assumptions.
7. The algorithms run in polynomial time as well.
8. They claim to have addressed the comments that they received in their previous submission.

Weaknesses:
1. The methods are simple, which is a good thing, as I said. They seem fairly standard though, so in those terms, the algorithms don't seem super novel.
2. Assumption 3 looks very strong to me. How realistic is that?

Other comments/questions:
1. In lines 66-68, they say that the mean estimation papers of Kamath et al. use $\ell_\infty$ concentration assumptions. Is that really true? The concentration happens in $\ell_2$ norm for both papers, I think. Like think of a standard, high-dimensional Gaussian. The decay is the same in all directions, so the distribution can be visualised as an $\ell_2$ ball. Maybe, I'm missing something?
2. In equation (2), $S_u \sim P_u^m$ instead of $P_u$?
3. It will be better to say what the randomness is over in Assumption 4.
4. Line 198 should end with "minimize".
5. Line 295 should read as "Furthermore, our results still hold...".

**Time Spent Reviewing:**

4

---

> ### Author Response · Authors · 2021-08-10
> **Response to review**
>
> Thank you for your time and constructive feedback. We will incorporate all suggestions on improving the presentation in the updated version. We now address detailed technical comments below:
>
> - **[Regarding Assumption 3 (smoothness assumption)]**: Assumption 3 is fairly standard in the privacy literature (e.g. (Bassily et al., 2013) or (Feldman et al., 2019, 2020)) or broader optimization literature (e.g. (Allen-Zhu & Hazan, 2016; Defazio et al., 2014)) and is often satisfied in practice. For example, consider logistic regression (or most generalized linear models) with bounded data and a bounded parameter set: the individual loss functions are smooth (with dimension-independent smoothness constant). Also note that the results only require $O(n^{3/2})$-smoothness, so in the regime of large $n$, our results directly apply to non-smooth losses by smoothing the individual losses (e.g., with the Moreau-Yosida transform see e.g. (Guzmán & Nemirovski, 2015)). As such, we do not believe that this is an overly restrictive assumption.
> - **[Regarding $\ell_\infty$ concentration in (Kamath et al, 2019) and $\ell_2$ concentration in our paper]**: This is a good point. As you mentioned, Kamath et al. use $\ell_\infty$ concentration assumptions, which also implies concentration in $\ell_2$. However, concentration in $\ell_2$ with radius $\tau$ only implies concentration with radius $\tau$ in $\ell_\infty$. Hence, naively applying the technique in Kamath et al will incur an additional $\sqrt{d}$ factor in the utility guarantee. We resolve this issue using a random rotation trick, which makes the dependence on $d$ only logarithmic.
>
> *References not in the paper:*
> - (Allen-Zhu & Hazan, 2016) Variance-reduction for faster non-convex optimization. Zeyuan Allen-Zhu and Elad Hazan. ICML 2016.
> - (Defazio et al., 2014) SAGA: A fast incremental gradient method with support for non-strongly convex composite objectives. Aaron Defazio, Francis Bach and Simon Lacoste-Julien. NeurIPS 2014.

---

> > ### Comment · Reviewer_GqWK · 2021-08-10
> > **Response.**
> >
> > Thanks for your response! Will update my review accordingly.

---

### Official Review · Reviewer_Vtrv · 2021-07-16

**Rating:** 7
**Confidence:** 4

**Summary:**

This paper deals with DP in a setting where all users contribute m datapoints to the given dataset, and the curator wishes to maintain DP on a user-level. The main result of the paper is a SGD algorithm (or an iterative aggregator) in which for each user the m datapoints are averaged then an update step takes place w.r.t the avg. Utility bounds for the approach are given in Thm 3 and 4. In addition the authors give a (close) lower bound in Thm 5.

**Limitations And Societal Impact:**

Irrelevant.

**Main Review:**

On the one hand, I really like the idea of DP for user-level privacy. Moreover, the "crux" of the paper is the clever observation that averaging the m datapoints of the user reduces the variance in the mean to about 1/\sqrt m, so less noise is needed under assumption A2 (and presumably also under A1). So, there's true novel observation in the paper.

On the other hand, I am uncertain whether the assumption that each user contributes *exactly* m points sampled i.i.d (or close to i.i.d) isn't too restrictive in some sense. I guess one can always average the m_i datapoints contributed by the user and check whether those are (roughly) within distance of 1/\sqrt{m_i} of one another (and if not - replace the avg with zero) and add noise proportional to 1/\epsilon\sqrt{m_i} --- it's just that the utility analysis gets hairier (presumably).

But all in all, this is a solid contribution on an interesting problem, so I vote for acceptance.

Some wording comments:
*Line 41: you mean - in the central model of DP?
* Lines 54-8: I'd be very careful in arguing anything about how the "real world" behaves. Some rewording is in order, especially since the idea here doesn't really come across clearly.
* Alg2: Each x_i \in [-B,B]^d (not ^n)

**Time Spent Reviewing:**

Refuse to time myself.

---

> ### Author Response · Authors · 2021-08-10
> **Response to review**
>
> Thank you for your time and positive feedback; we will address the comments on the presentation and wording in the revised version of the paper. Regarding the question of **different users contributing different numbers of samples**, our algorithms directly extend to this setting by just replacing $m$ (in the utility bounds) with $\mathsf{median}(m_1, \ldots, m_n)$. In the case where the median is very small (or a constant), extending our results is a very interesting avenue for future work indeed.

---

> > ### Comment · Reviewer_Vtrv · 2021-08-26
> > **Acknowledge the authors' respose.**
> >
> > Review remains unaltered.

---

### Official Review · Reviewer_7Pkz · 2021-07-19

**Rating:** 6
**Confidence:** 3

**Summary:**

The paper proposes a new private mean estimation mechanism with estimation error proportional to the concentration radius of queries. The mechanism is applied to stochastic optimization algorithms to solve user-level private empirical risk minimization (ERM) and user-level private stochastic convex optimization (SCO) with improved performance for concentrated queries.
Contributions:
1. A new private mean estimation mechanism with error scaling with query concentration radius.
2. New algorithms for user-level private ERM and user-level private SCO with error scaling with concentration radius of gradients.

**Limitations And Societal Impact:**

Yes.

**Main Review:**

The core of the paper is a private mean estimation algorithm with error scaling with concentration radius instead of problem dimension. By applying the mean estimation mechanism in SGD based optimization algorithms, one can solve user-level private ERM and user-level
 private SCO with error scaling with gradient concentration radius. Overall the contribution of all the results combined seems enough for a publication.

Strengths:
1. A new private mean estimation is proposed with good utility guarantees.
2. New private optimization algorithms with good utility guarantees for concentrated gradients.

Weaknesses:
1. For the limited heterogeneity setting, the maximum degree of allowed heterogeneity is not very clear. More specifically, it might be better to explicitly write down the maximum possible rate of \Delta in Corollary 1, Theorem 3, etc. Or the authors could put \Delta in the bounds to make it more clear. More discussion should be put on heterogeneous settings in these bounds since for user-level DP, it is too optimistic to assume homogeneous distributions.
2. It seems user-level DP in the homogeneous setting does not have much difference from sample-level DP.

-----------------after rebuttal---------------------

Thanks for the response, my concerns are addressed.


**Time Spent Reviewing:**

2

---

> ### Author Response · Authors · 2021-08-10
> **Response to review**
>
> Thank you for your time and constructive feedback. We now address the detailed comments below.
>
> - **[Regarding the assumption on heterogeneity]**: Thank you for the suggestion; we will add this in the revised version. We would like to point out two things: First, we provide a “user-friendly” result in Appendix C showing how to handle the heterogeneous setting for arbitrary statistical problems, expressing the excess error incurred in this case as a function of the heterogeneity $\Delta$. Second, as we note in the paper, our results also hint at concrete recommendations on how to collect new samples under any heterogeneity level: increasing $m$ will yield the most value in the i.i.d. setting, and will yield no improvement when the users’ distributions are arbitrary. As the real-world will lie somewhere in between, our results exhibit a regime where, regardless of the level of heterogeneity, it is better to collect more users (increasing $n$) than more samples (increasing $m$). Characterizing these rates under various and more general heterogeneity assumptions is of course still a very interesting and important open problem.
> - **[Comparison between user-level DP and sample-level DP in the homogeneous setting]**: User-level DP is a more stringent notion than sample-level DP as it requires that the output of the learning algorithm should not change much if any number of data points in the user's dataset changes (i.e., up to their whole contribution). Hence, the utility-privacy trade-off is different, as characterized in our work.
> In terms of techniques, any sample-level DP algorithm can be used to provide meaningful guarantees under user-level DP by applying group property of DP. However, to get optimal rates, we need fundamentally different techniques as we show in the paper. We hope this answers the question. Let us know if the reviewer is referring to other comparisons between user-level DP and sample-level DP.

---

### Official Review · Reviewer_QMQQ · 2021-07-20

**Rating:** 7
**Confidence:** 4

**Summary:**

The paper studies learning problems under so-called user-level differential privacy: in this setting, each one of n users holds m data points of the training set. This type of privacy then requires that under a change of *any number of data points belonging to a single user*, the distributions over the outputs of the algorithm are close in the sense of (epsilon, delta)-DP. A trivial way this could be achieved is group-privacy, where the privacy parameters would degrade linearly with the number of samples m.

The paper first gives a private mean estimation algorithm for the 1-dimensional case, which, given an input dataset that is both bounded in [-B,B] and (tau, gamma)-concentrated (that is, with probability gamma all points are within tau of a center point), returns an estimate that is close to the empirical mean plus a Laplace noise term of order tau/epsilon*n. This is then extended to the d-dimensional setting, using a random rotation of the data to ensure that each dimension is bounded appropriately in ell-2 norm rather than the ell-infty norm, and using the 1-dimensional solution on each dimension. Both these algorithms are *not* user-level DP but are used in the rest of the paper as building blocks of user-level DP algorithms.

The first application of these is a user-level DP d-dimensional mean estimation algorithm which, given samples from a distribution P supported on the euclidean ball with radius B, has squared ell-2 error in the order of sqrt{Var(P)/(m*n)}+\sqrt{d}*B/(\sqrt{m}*n*epsilon). The error due to privacy in this result degrades with 1/\sqrt{m} rather than being independent of m which is what the trivial approach would give us.

Using these tools, the authors then present user-level DP algorithms for empirical risk minimization with smooth convex, strongly convex, and non-convex losses as well as for stochastic convex optimization (SCO) with subgaussian gradients, demonstrating this improvement of 1/sqrt{m} in the error bounds.

They finally prove lower bounds that show that both the latter result on SCO (although not in all generality) as well as the d-dimensional user-level DP mean estimation bound are tight.


**Limitations And Societal Impact:**

Sufficiently discussed.

**Main Review:**

Strengths:
- I think that the paper is very well motivated by practical applications and studying the effect of multiple user entries is meaningful.
- The paper includes a variety of results for learning problems under user-level DP (even slightly more than described above) which makes it a useful extensive study of this notion.
- It is generally well written.

Weaknesses:
- It appears that the techniques used in the paper are not particularly surprising, as the mean estimation algorithms which are used as building blocks for the rest of the results are mostly based on the known approach of Karwa-Vadhan, and the way this is extended to the user-level DP algorithm is by taking the average of the contributions of each user and applying the algorithm above. The random projection trick in the d-dimensional mean estimation case is nice. That said, this collection of results requires detailed work and I find that the results themselves are valuable enough.

*Detailed comments/typos:*
- How necessary is the bound B in the mean estimation result? The Gaussian mean estimation of Karwa-Vadhan (and in general the stable-histogram algorithm) works even for infinite number of bins using (epsilon,delta)-DP. Would it be possible to state your results more generally?
- I understand that the TV-closeness statements help in the results about in the limited heterogeneity setting, but I would have found a statement of the form “w.p. 1-beta, the error is…” easier to parse. Perhaps adding these after the theorem statements might help.
- line 35: I actually found the phrase “does not significantly change if any single element of a user’s data changes” in the intro confusing. If we know that only one data point changes then neighboring datasets satisfy the classic definition of adjacency that we have in DP. It is made clear by the previous and later more formal definitions but perhaps you might want to revise this particular phrase.
- line 198: wishes to minimizes -> wishes to minimize
- Algorithm+Theorem 2: Shouldn’t both alpha (the desired failure probability) and gamma (the concentration failure probability) be inputs to the algorithm? The algorithm takes only gamma and the theorem only alpha.
- line 234: Z in the expectation is not defined close to the theorem.
- Minor: The use of mu for mean and strongly convexity and the use of H for the Hadamard matrix and smoothness parameter could maybe be avoided.
- Algorithm 3+Corollary 2: Again alpha is set but is not used in the algorithm as an input.
-line 322: lossesof -> losses of
-line 754: In Lemma 2, the maximum on the LHS is over i but the RHS does not have a quantifier (which i is it for?). I think it should be replaced by a “for all i: ||Uz_i||_infty<=…||z_i||_2…”
-line 763: delta->alpha

===================

Thank you for the response --  I still recommend acceptance.

**Time Spent Reviewing:**

8

---

> ### Author Response · Authors · 2021-08-10
> **Response to review**
>
> Thank you for your thorough and detailed feedback, we will address all of your comments regarding presentation in the final version. We now address main technical concerns raised by the reviewer.
>
> - **[Regarding bound B in the mean estimation result]**: We believe our results extend to the setting where the r.v. is not bounded but there is a (crude) bound on the value of the mean (which would enter logarithmically in the utility analysis). Further extending it to unbounded domains is an interesting direction to explore but we have not pursued results at this level of generality.
> - **[Regarding TV-closeness statement]**: Thank you for this suggestion, we will fix this in the updated version; note that as is, these can be directly deduced with a tail bound on Laplace noise. We would like to point out that mean estimation is mostly a tool for solving ERM and SCO; for these, we provide high probability and in-expectation utility bounds, respectively.
> - **[Informal definition of line 35]**: Thank you also for pointing this out, we will fix it and provide better intuition.
> - **[Regarding Theorem+Algorithm 2]**: You are correct, both $\alpha$ and $\gamma$ should be inputs to the algorithm (and the theorem); we will rectify this in the revised version of the paper.

---

### Decision · Program_Chairs · 2021-09-27

**Decision:**

Accept (Poster)

**Comment:**

This paper deals with DP in a setting where all users contribute m datapoints to the given dataset, and the curator wishes to maintain DP on a user-level. The main result of the paper is a SGD algorithm (or an iterative aggregator) in which for each user the m datapoints are averaged before an update step takes place w.r.t the avg. The reviewers agree that this is an interesting paper with solid contribution.